# Amber rainbow ribbon effect in broadband optical metamaterials

Jing Zhao [1,3] ✉, Xianfeng Wu [2,3], Doudou Zhang[2], Xiaoting Xu[2], Xiaonong Wang[2] & Xiaopeng Zhao [2] ✉

Using the trapped rainbow effect to slow down or even stop light has been widely studied. However, high loss and energy leakage severely limited the development of rainbow devices. Here, we observed the negative Goos-Hänchen effect in film samples across the entire visible spectrum. We also discovered an amber rainbow ribbon and an optical black hole due to perfect back reflection in optical waveguides, where little light leaks out. Not only does the amber rainbow ribbon effect show an automatic frequency selection response, as predicted by single frequency theoretical models and confirmed by experiments, it also shows spatial periodic regulation, resulting from broadband omnidirectional visible metamaterials prepared by disordered assembly systems. This broadband light trapping system could play a crucial role in the fields of optical storage and information processing when being used to construct ultra-compact modulators and other tunable devices.

Slow light with a remarkably low group velocity offers an opportunity to manipulate light propagation. Slow-wave structures can slow or even stop electromagnetic waves, suitable for enhancing light–matter interactions and flexibly controlling optical flow[1,2]. Slowing down light has generated considerable interests for developing all-optical nonlinear devices[3,4], optical data storage[5], optical modulators[6], switches[7], buffers and delays lines[8,9], and other miniaturized photonic devices. Among several promising methods to slow down light, the most prominent technique so far is based on electromagnetic induced transparency (EIT) from quantum interferences[10,11]. However, major challenges, such as non-solid-state devices and harsh experimental conditions, are to be overcome before quantum EIT could be made for practical applications.

The trapped rainbow effect is a unique method to achieve light slowing down, which is demonstrated theoretically by Hess et al.[12]. This effect is induced by the negative Goos–Hänchen (GH) shift in slowly, spatially varying negative-index heterostructures, resulting in a broadband light beam scattering into different frequency-specific locations. It is thus a promising candidate for creating practical optical metamaterials[13,14], as well as plasmonic structures[15–18] and PCs[19–22]. Previously, trapped rainbows at visible frequencies were experimentally observed in dendritic cell cluster metasurface waveguides[13,14], but those

systems rely on metallic structures that are vulnerable to intrinsic Ohmic loss. To address this issue, all-dielectric taper[23] based on Mie resonance surface mode was later developed to achieve broadband light trapping at microwave frequencies. In addition, a gradient metasurface was designed at microwave bands to realize rainbow trapping, in which the resonant-enhanced absorption of split-ring resonator basic cell was used to suppress reflections[24]. Tunable rainbow capture was obtained by establishing high localized field strengths, using trench width and trench length as tuning parameters of plasma devices[25]. Similarly, some studies have predicted and experimentally demonstrated the acoustic rainbow trapping and field enhancement effects[26,27]. Tapered magnetic fields in nonreciprocal waveguides has been proposed to achieve rainbow trapping with fully stopped light[28]. These designs involved the use of gyromagnetic materials[29] or an external magnetic field[30], significantly increasing system complexity. Recently, some advances were made toward topological rainbow effects[31–37] based on the topological photonics theory via controlling the group velocities of topological photonic states. Although much has been reported in the literature, trapped rainbow effect in the visible spectrum still faces many roadblocks on its way of becoming a practically usable technology. These include great difficulties in preparation, high resistive heating loss, and the long-term storage of light, to name a few.

[1]Medtronic Plc, Boulder, CO 80301, USA. [2]Smart Materials Laboratory, Department of Applied Physics, Northwestern Polytechnical University, Xi'an 710129, P. R. China. [3]These authors contributed equally: Jing Zhao, Xianfeng Wu. ✉e-mail: zhaojing1120@gmail.com; xpzhao@nwpu.edu.cn

It is widely known that optical negative index metamaterials suffer from narrow bandwidth and large propagation losses[38–40]. Metamaterials that are made of sophisticated 3D networks are typically difficult to fabricate using conventional monolithic lithography[41], which is complex, expensive and time-consuming. Broadband and omnidirectional visible spectral metamaterials are even more challenging to create. Any imperfection in preparation would affect materials properties that directly contribute to light slow down. Recently, we made significant progress in optical metamaterials. The high loss issue was resolved by a ball-thorn-shaped meta-clusters with symmetrical structure consisting of dielectrics coated by a super-thin silver layer[42]. Furthermore, based on the weak interaction[43–45] between ultra-low loss isotropic meta-cluster systems, the disordered self-assembly of wideband omnidirectional metamaterials with different scale cluster structures is proposed to break through the bottleneck of three-dimensional structure preparation[46]. Our partial disorder effect actually relies on disordered hyperuniform media of recent years, and related order metrics[47,48], and these provide the core of broadband properties with low dependence on the incidence angle. Our work[42,46] overcome the difficulties of high loss, broadband and orientation in the field of metamaterials, which can achieve negative refraction in the visible spectrum and exhibit frequency adaptive response. The great progress makes it convenient to assemble optical metamaterials and optical devices from the bottom-up[49,50].

In this work, we constructed heterogeneous axial film waveguides based on above mentioned broadband omnidirectional visible metamaterials and verified their ability to slow down light. These broadband planar film samples were assembled by a list of narrowband, ultra-low loss isotropic meta-cluster systems without purposeful selection manually. We observed negative GH shifts in these samples ranging from green light to red light. In addition, using a broadband waveguide consisting of two such planar samples, we observed an amber rainbow ribbon and an optical black hole, in which the light is nearly stopped. This result is a piece of direct evidence of broadband light trapping compared to other slow light systems. It suggests that this type of metamaterials prepared by disorderly assembled cluster systems could respond to a broadband external stimulation in a self-selected manner both in frequency and in space. This work is the latest demonstration of possible applications of ultralow loss broadband omnidirectional negative refractive index metamaterials. In particular, the broadband light trapping system could enable exciting developments in future ultra-compact optical modulators and other tunable optical devices.

## Results

### The waveguides of broadband planar film samples

Figure 1a shows the schematic diagram of the construction of axially varying heterogeneous film waveguides for slowing down light, based on proposed broadband omnidirectional visible metamaterials. In order to achieve low losses, the ball-thorn-shaped cluster model is applied, and theoretical calculations show that this meta-cluster reduces the thickness of the metallic silver layer required to generate local plasmon resonance (LPR) from the micrometer scale to the height of two or three atomic layers. The experiment further shown that plasmonic resonance can be achieved by replacing a continuously distributed silver layer with a discretely distributed metallic silver layer. We have achieved the experimental determination of negative refraction in a three-dimensional visible metamaterial sample and confirmed the low loss of this material[42]. Using the monochrome particles red and green[42] and the combination of eight kinds of particles[46], monochrome film and broadband film can be prepared respectively (see "Methods"). The broadband planar film samples are assembled automatically by a list of narrowband, ultra-low loss isotropic meta-cluster system without selection, covering the main bands

from green light to red light. The broadband waveguide consists of two planar samples (see "Methods").

As has been reported, we have invented an ultralow loss isotropic metamaterial in the visible spectrum[42], and the ball-thorn-shaped metamaterial cluster model consist of a spherical kernel and many protruding rods (Fig. 1b, c shows green light and red light models). In the simulation, both the kernel and rods are made of $TiO_2$ coated by Ag of 1 nm in thickness. In total, 600 identical rods with cross-sectional diameter of 15 nm are uniformly distributed around the surface of a kernel. $l$ represents the diameter of the meta-cluster, $r$ is the radius of the spherical kernel, and $P$ refers to the lattice constant of the meta-cluster, the meta-cluster is fully immersed in polymethyl methacrylate (PMMA). The structural parameters of red-light model are $l = 640$ nm, $r = 215$ nm, and $P = 670$ nm, and that of green-light model are $l = 530$ nm, $r = 165$ nm, and $P = 560$ nm. Adjusting $l$, $r$, and $P$ according to scaling rule enables the model to be applicable to different working frequency bands. The cluster is named $Ag/AgCl/TiO_2@PMMA$, and the ball-thorn-shaped meta-clusters with symmetrical structure consisting of the dielectric and its surface dispersed super-thin silver layer have replaced the lithographically defined meta-atoms in existing negative-index metamaterials (NIMs); it is found that the discrete super-thin silver layer produced by the photoreduction method can generate plasmon resonance when excited by electromagnetic waves, thereby achieving the performance of metamaterials. The significant reduction in silver coating thickness provides the physical basis for the decreased joule heating and the realization of ultralow losses. Subsequently, we successfully obtained a broadband omnidirectional metamaterial (Fig. 1d shows the unit of broadband omnidirectional meta-clusters system) randomly assembled by a list of narrowband, omnidirectional, and ultralow loss meta-cluster system using a bottom-up approach[46]. Figure 1e shows the transmission electron microscopy (TEM) images of the particles that resonate in the green (left) and red (right) light spectrum, revealing a classic kernel–shell structure. The TEM images show that the size of the ball-thorn-shaped $Ag/AgCl/TiO_2$ particle is ~500–700 nm, and the thickness of the PMMA shell is nearly 20–30 nm. The negative refraction for red sample occurs at around 610–640 nm, and the minimum refractive index is −0.41 at 630 nm; the negative refraction for green sample occurs at around 520–550 nm, and the minimum refractive index is about −0.30 at 532 nm. Figure 1f shows the scanning electron microscopy (SEM) image of broadband film sample experimentally obtained by disordered self-assembly of eight kinds of single-frequency meta-clusters. The negative refractive index of 490 nm–730 nm band (Fig. 1g) and broadband inverse Doppler effect across most of the visible spectrum was observed in the proposed broadband metamaterials. The lowest figure of merit (FOM = −Re(n)/Im(n)) was 6.7 at 538 nm and the highest value was 13 at 592 nm, which predicts low losses[46]. Simulated refractive index of the green and red metamaterials and approximations of the real part of the refractive index calculated with the Kramers–Kronig integral[51] are shown in Fig. 1h, and they basically satisfy the KK relationship. In addition, related study[1] has shown that that the negativity in the real part of the effective refractive index enables the deceleration of light in the layers of the heterostructure. On these bases, the research in this paper can be conveniently carried out.

### Measurement of Goos-Hänchen (GH) shift

The study of Hess et al.[12] indicated that incident light with different frequencies will be trapped at different critical thicknesses in an axially varying heterostructure, due to the negative GH effect. Therefore, the ability of metamaterials to achieve negative GH effects is the basis of rainbow trapping experiments.

Many studies have been conducted over the years due to the importance and interest of the GH effect and the trapped rainbow effect. For example, the work in ref. 14 was realized by a metallic dendritic meta-surface. Moreover, past work was only available for

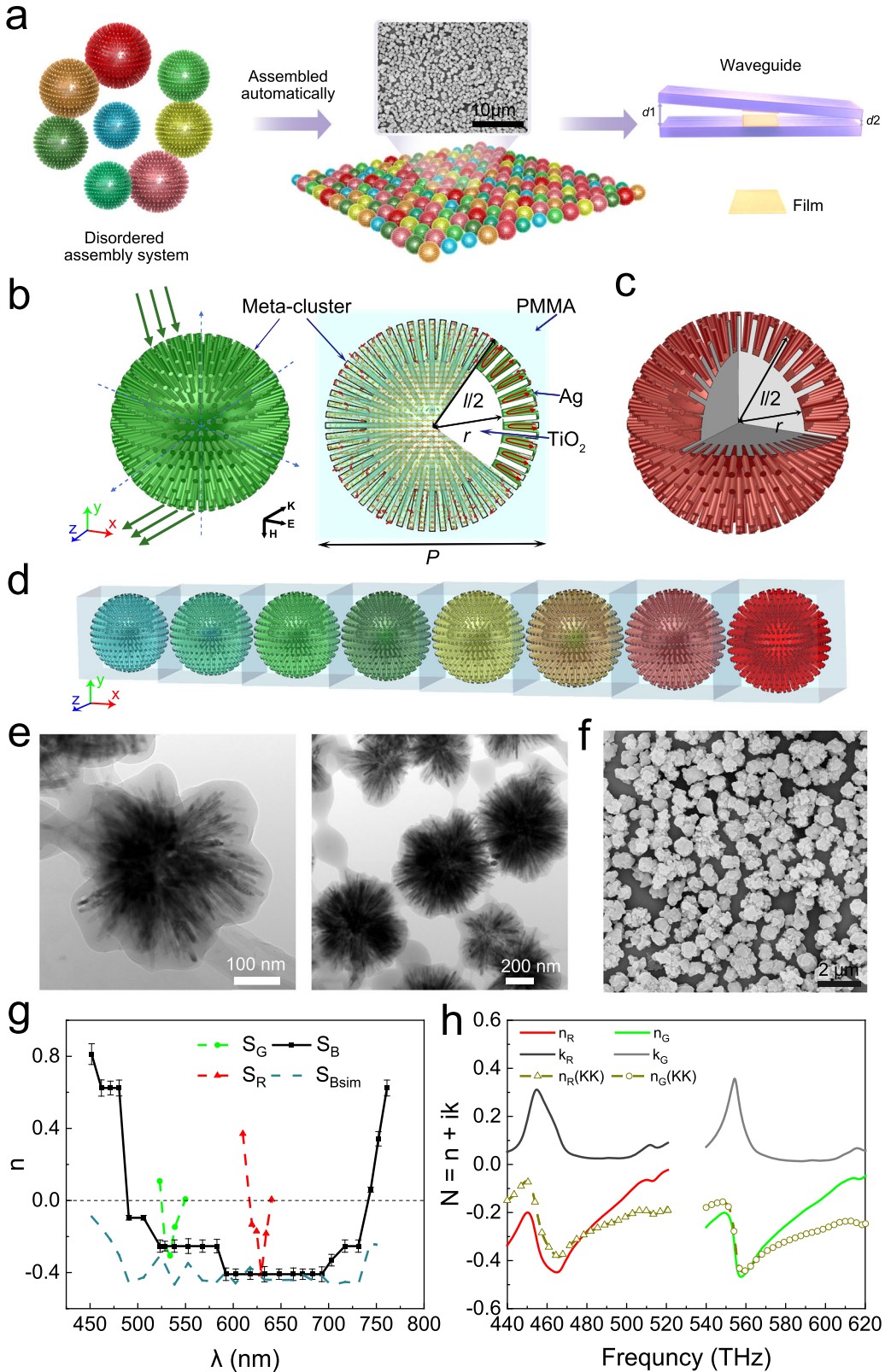

materials with narrow bands (as shown in Fig. 2c), i.e., a sample acts only in a specific band. Based on the theory of ref. 12, the trapped rainbow is generated by the negative GH effect of metamaterials with negative refractive index. However, the difficulty of obtaining visible spectrum materials with negative refractive index has constrained the progress of the research, making this study has not been completed in metamaterials. Recently, our group has designed and prepared ball-

thorn-shaped ultralow loss, omnidirectional, and broadband visible metamaterials, and experimentally determined the negative refractive index of the three-dimensional materials[42,46], which provides the basis for realizing this work.

Figure 2a shows the schematic of experimental setup for GH shift measurement, which is achieved by the interference of S-polarized and P-polarized beams[14,52,53]. The polarizer P1 is used to convert incident

**Fig. 1 | Conceptual illustration of the axially varying heterogeneous wave-guides based on proposed broadband omnidirectional visible metamaterials. a** Schematic diagram of axially varying heterogeneous waveguides composed of broadband planar film samples. **b** Green light meta-cluster model with parameters of $l = 530$ nm, $r = 165$ nm, and $P = 560$ nm, the cluster is composed of a spherical core and many prominent rods (left). When the response band light ray incident, the meta-clusters will occur negative refraction effect. The profile current distribution perpendicular to the external magnetic field and the 1/8 model profile (right). **c** Red-light meta-cluster model with parameters of $l = 640$ nm, $r = 215$ nm, and $P = 670$ nm. **d** Cluster unit of broadband omnidirectional meta-clusters system; it is composed of eight clusters with response bands of 490, 500, 540, 570, 600,

640, 680, and 700 nm, respectively. **e** TEM images of green-light (left) and red-light (right) particles. The size of the ball-thorn-shaped Ag/AgCl/TiO$_2$ particle is ~500–700 nm, and the thickness of the PMMA shell is nearly 20–30 nm. **f** SEM image of the monolayer film of the broadband metamaterial, which is obtained by disordered self-assembly of eight kinds of single-frequency particles. **g** Refractive index curve measured for broadband sample S$_B$, green-light sample S$_G$, and red-light sample S$_R$. S$_{Bsim}$ is the simulated value of the broadband sample. **h** Simulated refractive index of the green and red metamaterials and approximations of the real part of the refractive index calculated with the Kramers–Kronig integral. **b**, **e** are cited from ref. 42; **f**, **g** are cited from ref. 46. Error bars represent the standard deviation of three independent experiments.

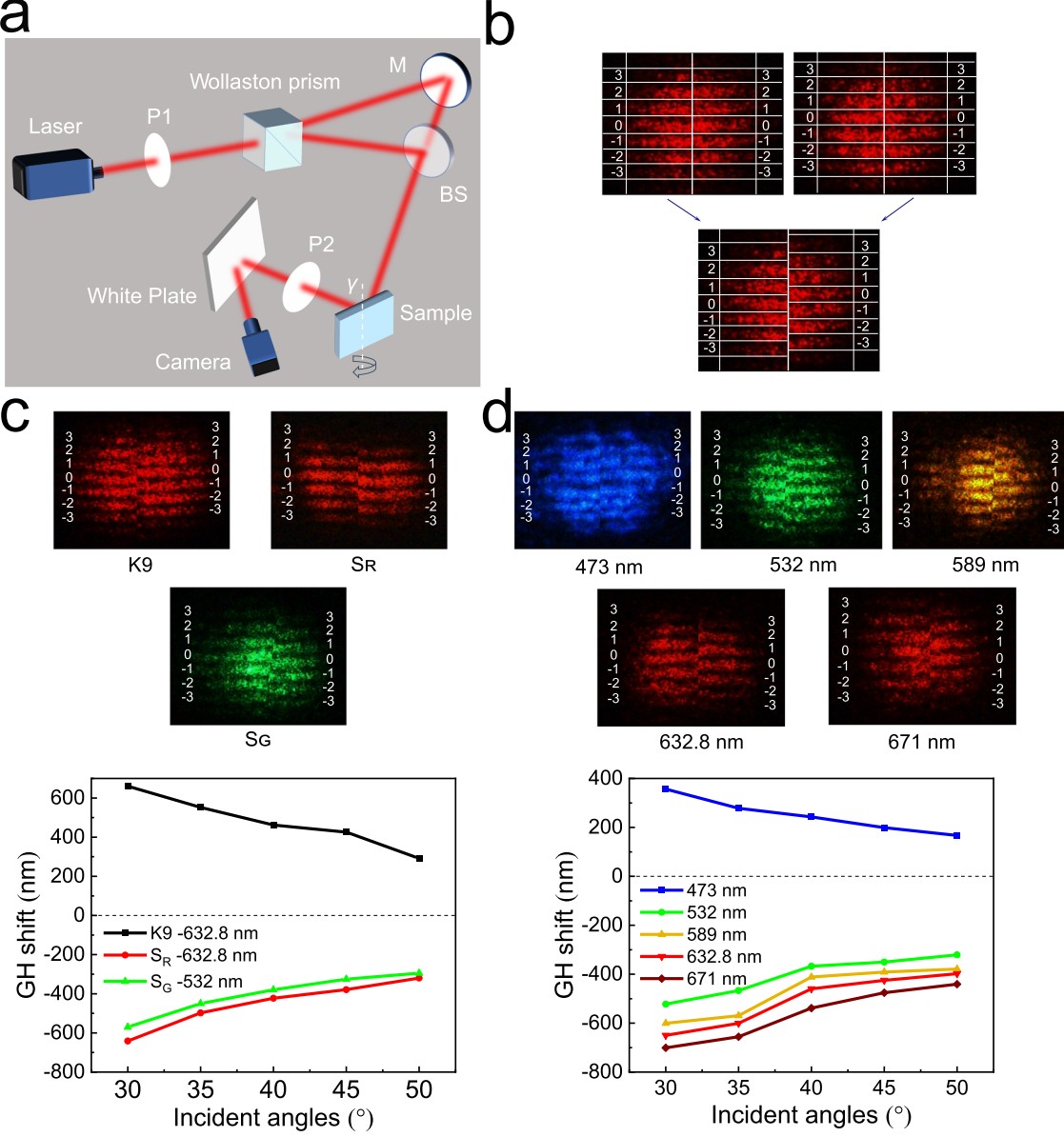

**Fig. 2 | Measurement of the GH shifts. a** Schematic of the measurement of the GH shift. M: mirror, BS: beam splitter. $\gamma$: incident angle. **b** Interference patterns are obtained when the polarization angles of the analyzer P2 are 45° (Upper left) and 135° (Upper right). Then the two patterns are horizontally spliced into an image (bottom) to calculate the phase difference $\Delta\varphi$. **c** GH displacement behavior of the

K9 crystal, red-light sample (S$_R$), and green-light sample (S$_G$) as a function of wavelength and incident angle. **d** GH displacement behavior of the broadband sample as a function of wavelength and incident angle. Upper panel: the spliced interference patterns.

light into linearly polarized light, and then the Wollaston prism is adjusted to split the linearly polarized light horizontally into p-polarized light and s-polarized light at a separation angle of ±15°. The p-polarized beam is reflected by the mirror M and reaches the BS, whereas the s-polarized beam reaches the BS directly. The two polarized beams coincide on the front surface and then propagate together. The overlapping beams incident on the surface of the sample at an angle of γ, and GH shift occurs when they are reflected by the sample. Afterward, the reflected beam passes through the polarization analyzer P2; an interference pattern appears on the white plate and is recorded by the CCD camera in real time. When the polarization angle of P2 is 45° or 135° with respect to the p- and s-polarized beams, interference fringes are generated by the s- and p-polarized beams. At other polarization angles of P2, no interference fringes are noticeable. The intensity of two orthogonal beam components can be fine-tuned by rotating P1, thereby effectively improving the sharpness of the fringes. The number of fringes is maintained at approximately seven in experiments to ensure accuracy. The interference fringes from 45° and 135° P2 are recorded (Fig. 2b, upper panel). Each level of the interference fringes is marked to observe the interference fringes and the moving direction of the GH shift. The brightest interference fringe is labeled 0, and the upward and downward interference fringes are labeled ±1, ±2, and ±3. Then, the two patterns are horizontally spliced into an image (Fig. 2b, lower panel). Given the difficulty of directly observing the moving direction and distance of the fringes, the pieced interferogram are imported into the PS software for processing to calculate the phase difference Δφ; thus, we can calculate GH shifts. During the experiment, the sample is affixed to a 360° rotatable platform with a clamp. The GH shift at different incident angles can be measured by rotating the platform. Finally, the GH shift is calculated using the following formula[52,53]:

$$\Delta x_p = \frac{\lambda}{2\pi} \Delta\varphi \frac{1}{\sin\gamma} \qquad (1)$$

where γ is the incident angle, Δφ is the change amount of the phase reflected by the movement of the interference fringe, and λ is the incident wavelength. Δx_p consists of the direction and magnitude of the GH shift.

GH shift of a standard K9 crystal is measured using the He-Ne laser with a frequency of 632.8 nm to verify the reliability of the optical path (Fig. 2c). The spliced interference patterns show that the interference fringes in the right part have an upward movement with respect to those in the left part, and a normal GH shift occurs after light passes through the K9 crystal. This result is consistent with those given in the literature[52] and verifies the effectiveness of our measuring optical path. Further measurements of the GH shifts for the red-light sample (S_R) and green-light sample (S_G)[42] at different incident angles (vary from 30° to 50°) are exhibited in Fig. 2c. The movement of interference fringes is opposite to that of K9, and GH shifts generated by S_R and S_G at corresponding resonance wavelength are all negative. Next, we measured the GH shifts of broadband metamaterial samples assembled by Ag/AgCl/TiO₂@PMMA particles[46]. It can be seen in Fig. 2d that negative shifts indeed occur when the wavelength of incident light beams matches the negative refraction wavelength (532, 589, 632.8 and 671 nm) of the planar samples, and a normal GH shift occurs at 473 nm (beyond the negative refraction wave band).

## Trapped rainbow effect

Slow-wave structures can be applied to optical cache and deep subwavelength optical waveguide[1,12,37]. In 2007, Hess et al. proposed a three-layer wedge-shaped optical waveguide model for stopping light waves. The middle layer of this theoretical model is filled by homogenous isotropic negative-index metamaterials, and both the top and bottom layer are made of ordinary positive-index materials. Spatial

separation of light waves at different wavelengths—"the trapped rainbow effect"—is a result of the anomalous GH shift when a forward propagating wave hit upon the interface between the positive and negative index layers[1,12]. Nevertheless, to the best of our knowledge, a trapped rainbow effect experimentally achieved through negative-index metamaterials in the visible spectrum has not been reported yet. Herein, we realized the trapped rainbow effect in an experimental setting by using wedge-shaped visible-light metamaterial waveguides assembled by Ag/AgCl/TiO₂@PMMA particles[42,46].

A schematic of this experiment setting is exhibited in Fig. 3a. The light emitted by the xenon lamp is transferred to a beam of white light through the monochromator. A beam of parallel light is obtained after passing through a collimator, and then the spot decreases in size after passing through lens 1, lens 2, and an aperture. And a spot with a diameter of ~0.5 mm is finally obtained. The spot is incident along the centerline of the wedge-shaped waveguide, and the resulting image is recorded using a CCD camera connected with a computer.

**3D wedge-shaped optical waveguide consisting of 3D wedge-shaped sample and air.** First, a 3D wedge-shaped optical waveguide is fabricated by assembling a wedge-shaped metamaterial sample onto a glass substrate (Fig. 3a). Incident angle of the incoming light beam can be indirectly adjusted by varying θ, which represents the rotation of the glass substrate relative to the horizontal plane. θ is the key geometric parameter to control the spatial separation of visible light at different wavelengths. For the waveguide presented in Fig. 3a, no trapped rainbow is observed when θ = 0°. A trapped rainbow starts to be seen as θ increases progressively (see Fig. 3b, c). These results demonstrate that these 3D wedge-shaped optical waveguides consisting of 3D wedge-shaped sample and air can trap visible light. It is also found in the experiment that the trapped rainbow effect would quickly weaken before completely disappear as θ reaches and passes a critical angle, which agrees with the theoretical prediction. The spectral distributions of "trapped rainbow" phenomena produced by the 3D wedge-shaped sample (responding to the green light) are measured using the fiber spectrometer at different tilt angles. As special showcase in Fig. 3d–f, seven distinct colors of a trapped rainbow inside the green-light metamaterial waveguide (Fig. 3b) can be clearly seen by naked eyes, at θ = 3.8°, 8.8°, and 11.31°. The visible light of different frequency components stops at various thicknesses of the wedge-shaped sample waveguide, thereby forming a rainbow in the space.

Theoretical model of "trapped rainbow" based on homogenous isotropic negative-index metamaterials have been proposed previously in ref. 12 for some time, however, the preparation of isotropic metamaterial media has always been a challenge. Using our recently realized ball-thorn-shaped, isotropic negative-index visible spectrum metamaterial[42], the experimental results of "rainbow" for three dimensional structures are obtained in visible light, and the results here are consistent with the conclusions in ref. 12. In contrast to the oriented structures reported in the literature[1,3,15,16], this structure does not require the determination of a specific light incidence direction, which greatly facilitates the experimental measurement of the optical behavior. The experimental results of the anomalous GH effect of the material provide further evidence that the group velocity reduction of light waves is the physical basis for realizing slowing down light.

**3D wedge-shaped optical waveguide consisting of two planar samples.** We constructed a 3D wedge-shaped optical waveguide consisting of two planar samples coated with single frequency Ag/AgCl/TiO₂@PMMA particles[42] and TiO₂@PMMA particles (Fig. 4a, left panel). The heights of the incoming and outgoing ports are d1 and d2 (d1 > d2, and d2 is set as 0 here), respectively. The trapped rainbow experiment of the waveguide with different d1 is performed using the experimental optical path in Fig. 3a. In this experiment, d1 must be small to satisfy the adiabatic approximation condition in the

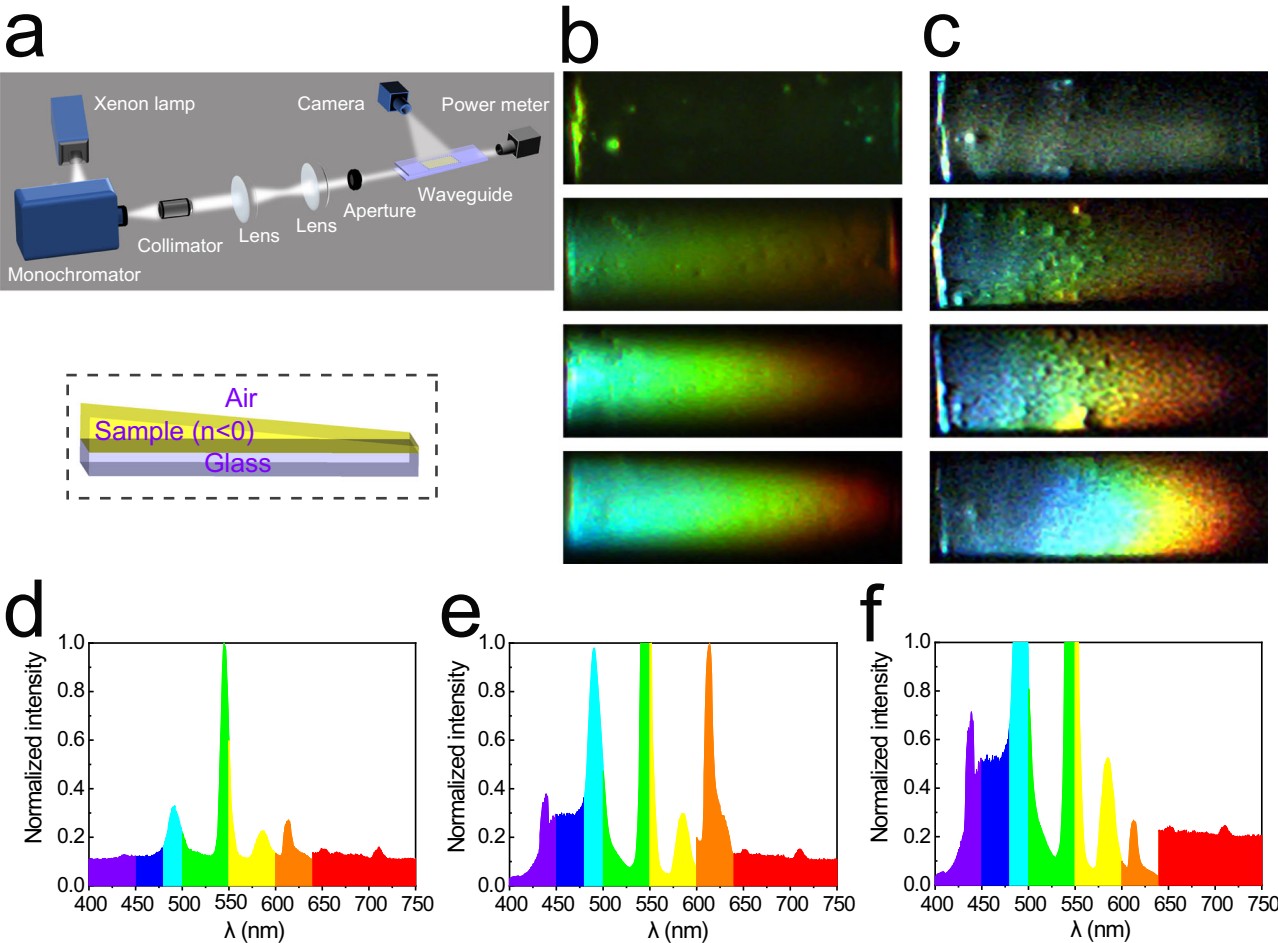

**Fig. 3 | Trapped rainbow effect in 3D wedge-shaped optical waveguide consisting of 3D wedge-shaped sample and air. a** Schematic diagram of the rainbow capture experiment device. Dashed box: schematic of a 3D wedge-shaped metamaterial waveguide. The wedge-shaped metamaterial layer (yellow) is sandwiched between the top air layer and the bottom glass layer (blue). The length of the wedge-shaped metamaterial layer with a wedge angle of about 2.3° is ~1 mm. A polychromatic light (white-light) beam enters the sample from the left-end port (the thicker side of the sample). $\theta$ is the angle between the glass substrate and the horizontal plane. **b** Photos of the trapped rainbow inside the green-light metamaterial waveguide with $\theta = 0°$, 3.8°, 8.8°, and 11.31° from top to bottom. **c** Photos of the trapped rainbow inside the red-light metamaterial waveguide with $\theta = 0°$, 4°, 6.38°, and 9.08° from top to bottom. **d–f** Visible light spectrum of a trapped rainbow captured inside the green-light metamaterial waveguide depicted in **b** with $\theta = 3.8°$, 8.8°, and 11.31°, revealing seven distinct colors.

theoretical model. A "trapped rainbow" phenomenon occurs when the thickness is adjusted to a suitable value[13,14]. The results of waveguide composed of green-light Ag/AgCl/TiO$_2$@PMMA particles are presented in Fig. 4b, and that of red-light Ag/AgCl/TiO$_2$@PMMA particles are presented in Fig. 4c. It is found that, for the waveguide composed of planar samples, the trapped rainbow effect only appears in the region where the negative-index metamaterials physically locate (Fig. 4b, c). Blue light appears on the side of the large port, and red light is close to the side of the small port. In particular, the low-frequency component (such as red light) of the white light stays in the thin part of the waveguide, and the high-frequency component (such as blue light) of the white light remains in the thick part of the waveguide. This condition is mainly due to the light of different frequencies differs in the abnormal GH shift that occurs at the interface between the metamaterial and the normal material. A set of controlled experiments verified that no trapped rainbow could be seen if only one planar metamaterial sample was in presence (Fig. 4d, e), and this results further indicate no other disturbance in the optical path.

Next, we constructed a wedge-shaped optical waveguide consisting of two planar broadband samples[46] coated with Ag/AgCl/TiO$_2$@PMMA particles (Fig. 4a, right panel), and the thickness of waveguide is adjustable by varying $d1$ (height of the incoming port). Experiments found

rainbow bands in the wedge-shaped waveguides of broadband samples. Figure 4f shows five "rainbow capture" states in which the opening of the incident port varies from 20 μm to 35.625 μm. It can be seen that the broadband waveguide exhibits significant differences from the narrowband green and red waveguides: I. A rainbow band appears. Single frequency particle system exhibits the rainbow effect, that is, each frequency component of the wave packet is stopped at a different guide thickness, leading to the spatial separation of its spectrum and the formation of a "trapped rainbow". Unlike the single frequency particle system, the broadband particle system shows a rainbow band phenomenon, i.e., the amber rainbow effect: each rainbow at a different location contains a variety of wavelengths of color. II. The amber rainbow band remains unchanged as the incident port changes. For red and green sample waveguides, the trapped rainbow effect would quickly weaken before completely disappear as the opening size reaches and passes a critical angle (Fig. 4b, c), which agrees with the theoretical prediction in ref. 12. But for the five states of Fig. 4f, the rainbow band remains when the angle of the incident light changes. III. Rainbow bands form periodic regulation. As can be seen from the above, the rainbow effect in monochromatic samples is similar to the theoretical model[1,12] and the reported experimental results[15–18], showing a frequency location correlation, that is an automatic frequency selection response. However,

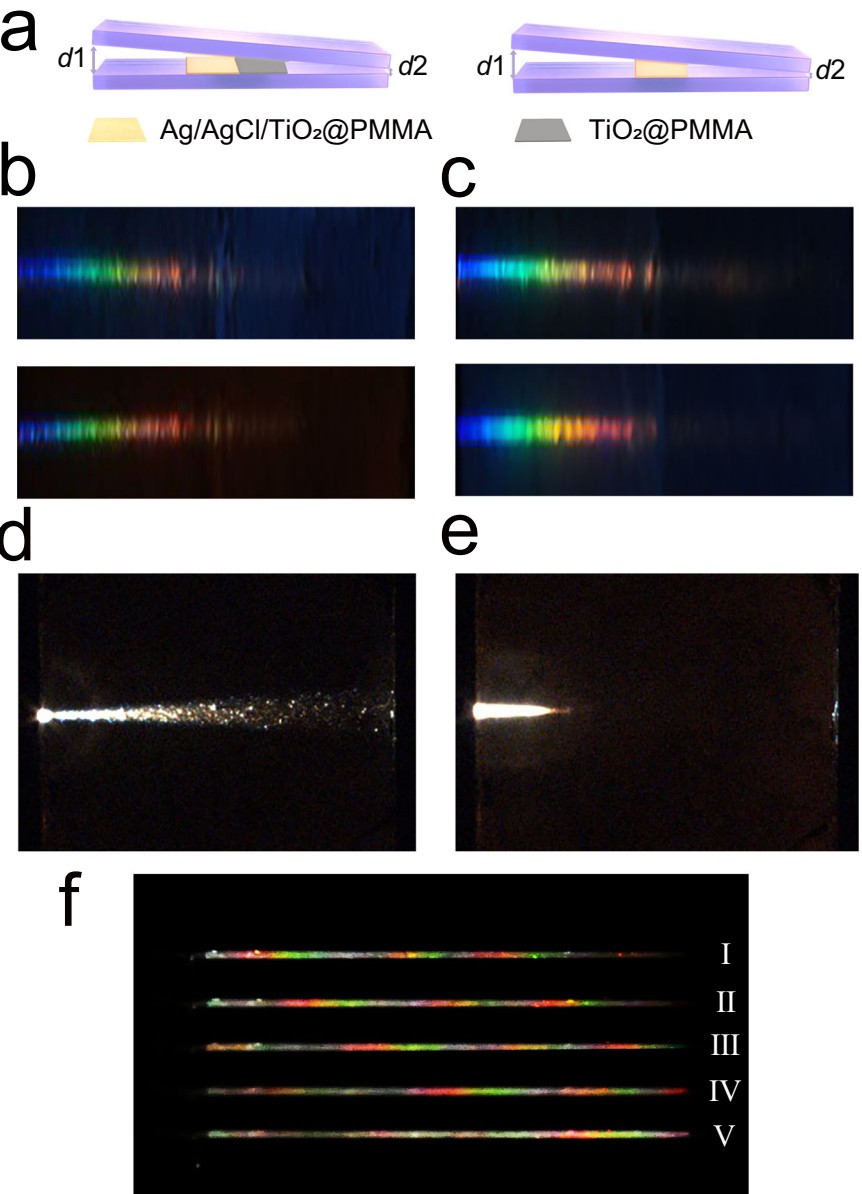

**Fig. 4 | Trapped rainbow effect of 3D wedge-shaped optical waveguide consisting of two planar samples. a** Schematic of a wedge-shaped metamaterial waveguide composed of two planar samples. Left panel: a 10 mm × 10 mm film composed of Ag/AgCl/TiO$_2$@PMMA particles (yellow part) and a 10 mm × 10 mm film composed of TiO$_2$@PMMA particles (gray part) are spin-coated on the left and right sides in the middle of the glass substrate, respectively. Right panel: a 10 mm × 10 mm film composed of Ag/AgCl/TiO$_2$@PMMA particles (yellow part) is spin-coated on the middle of the glass substrate. Hydrophilically treated glass of 50 mm × 10 mm are used as a substrate. The height of the incoming and outgoing ports is $d1$ and $d2$ ($d1 > d2$), respectively. The medium between the two planar samples is air. **b** Photos of the trapped rainbow inside the green-light metamaterial waveguide with $d1 = 138\,\mu m$ (top) and $d1 = 230\,\mu m$ (bottom). **c** Photos of the trapped rainbow inside the red-light metamaterial waveguide with $d1 = 138\,\mu m$ (top) and $d1 = 230\,\mu m$ (bottom). **d**, **e** Photos obtained when the tilt angles of the planar sample are 0° and 7.2°, respectively. **f** Rainbow capture image shows five "rainbow capture" states when the incident port $d1$ of the broadband sample wedge-shaped waveguide changing from 20 μm to 35.625 μm. I: $d1 = 20\,\mu m$, II: $d1 = 23.125\,\mu m$, III: $d1 = 26.25\,\mu m$, IV: $d1 = 29.375\,\mu m$, V: $d1 = 32.5\,\mu m$. A monolayer broadband sample film with an average thickness of ~0.7 μm and a size of 10 mm × 10 mm is deposited above and below the waveguide center.

the broadband sample showed the amber rainbow ribbon. Not only does the amber rainbow ribbon effect show an automatic frequency selection response, as predicted by single frequency theoretical models and confirmed by experiments, it also shows spatial periodic regulation, which is exactly the result of the broadband omnidirectional visible metamaterials that prepared by disordered assembly system. IV. Weak interaction properties. As in the case of negative refraction and inverse Doppler effect[42,46], the phenomenon of group velocity reduction in the broadband disordered particles system depends only on the physical properties of the particles and is not affected by the geometrical

distribution of the particles in the space, highlighting the weak interaction properties of the metamaterial structure.

The experimental results in Fig. 4 show that the trapped rainbow effect is based on the anomalous GH effect in negatively refractive media. The rainbow effect occurs in single metamaterials where the particles resonate at the same frequency, such as red and green metamaterials (Fig. 4b, c). For broadband structures consisting of multiple composite metamaterial particles, a rainbow band, i.e., the amber rainbow phenomenon, is formed (Fig. 4f). The physical origin of it is still the anomalous GH effect, where amber rainbow appears due to

the difference in resonance wavelengths of the different particles. As in the case of negative refraction appearances, broadband negative refraction results from the weak interactions exhibited by different particles. Here, each particle can produce an anomalous GH effect and appears as a corresponding rainbow, thus overall forming an amber rainbow. These results further illustrate that the "rainbow" phenomenon in the system that slows down the group velocity of light is the behavior of the metamaterial, which is controlled by the meta-cluster constituent units. At the same time, the units in the microstructure are randomly arranged, but do not affect the macroscopic overall behavior of the material. This behavior of the weak interaction system is quite different from strongly interacting systems such as photonic and acoustic crystals. This feature provides considerable convenience for designing metamaterial devices.

The theoretical model for slowing waves with homogenous isotropic negative index metamaterials presented the concept of trapped rainbow[12] in the infrared band, which exhibits diffraction of each wavelength at a different location, and has been approved by a considerable number of researchers[13–22]. This phenomenon in the visible wavelength band has been experimentally achieved at the waveguide interface using our three-dimensional negative index materials[42], which is more advanced than that reported in the literature for microwave wavelengths. Moreover, the adoption of a system of weakly interacting particles[46] at different scales allows the amber rainbow band to be observed. The experimental phenomenon is shaped like a rainbow in nature, which is more aptly described by the term "trapped rainbow". This spatial spectral separation is completely different from the prism dispersion model, and it is an intuitive, pictorial view of the "trapped rainbow" caused by the anomalous GH effect of metamaterial media, which is a direct proof of the theory that light with different frequencies is trapped at different locations in axially varying inhomogeneous waveguides composed of positive-negative refractive-index media. According to the theoretical model proposed by Hess et al.[12] and verified by many subsequent related works[13–22], this trapped rainbow phenomenon opened a way to slow down and even fix light over a wide frequency range. It offers important potential direct applications in the construction of nanophotonic wavelength routers, optical storage and optical buffer devices.

Overall, the aforementioned experimental results show that the metamaterial wedge shaped waveguide with negative GH effect shows the rainbow effect, in which light of different frequencies stopping at different positions; moreover, as the opening size of the waveguide decreases gradually, the rainbow shows a change from the short wavelength of blue to the long wavelength of red; when the opening size of the waveguide decreases to a certain degree, the phenomenon of zero energy flow occurs; the behaviors are the same regardless of whether the particles of the red or green wavelength, which is in line with the theoretical prediction of ref. 12. The experiments now further show the amber rainbows and the formation of black hole phenomena with zero energy flow in broadband metamaterial waveguide. This is the result of spatially periodic modulation of multiple rainbows by the disordered assembly metamaterials system and the generation of perfect back reflections, which greatly enhance light-matter interactions and exceeds the predictions of previous theories, revealing that the experimental results encompass a more complex and profound physical mechanism that needs to be further explored and will drive the development of a more profound theoretical system.

## Perfect back reflection: an optical black hole

The 3D wedge-shaped optical waveguides consisting of two planar samples mentioned above are also used for optical power measurement, respectively, and the heights of the incoming and outgoing ports are $d1$ and $d2$ ($d1 > d2$, and $d2$ is set as 2.4 μm here). We used the setup shown in Fig. 3a and measured the variation of the output optical power from the rear port of the waveguide with the size of the waveguide incoming port $d1$, at a known and constant incident power. The experiments were conducted in an optical darkroom, the effective optical power incident into the waveguide was limited to 31 pW and a NOVAII handheld optical power tester with measurement accuracy of 1 pW was used. Tables 1–3 show the test data for the output energy flow of the green, red and broadband materials as well as the TiO$_2$@PMMA waveguide, respectively, which were then normalized by taking the incident optical power as the maximum value (31 pW) to obtain Fig. 5a–c. We measured the variation of the outgoing optical power with the size of incoming port $d1$ of the waveguide, and the results show the state of the light in the waveguide. It can be found that for a fixed outlet $d2 = 2.4$ μm, the decreasing incident opening of the waveguide leads to variations in the outgoing optical power, and the light is completely stopped when reaching the critical thickness, and the emitted optical power decreases to 0. When the opening size is increased over a certain thickness, the trapped light escapes rapidly, and the outgoing optical power increases and tends to stabilize, and the waveguide is almost without the ability of the loss of light, at which time the loss of optical power is due to the inherent loss of the waveguide material (if there is no loss of light in the waveguide, the normalized output optical power should be 1).

The back reflection effect in the light-stopping state was observed experimentally. Figure 4a–c can illustrate that there is a rainbow effect in the thin film waveguide composed of Ag/AgCl/TiO$_2$@PMMA particles (yellow part), as a result of backward reflection due to the anomalous GH effect, while there is no anomalous GH effect and back reflection occurring in the waveguide composed of TiO$_2$@PMMA particles (gray part), and therefore it cannot cause the rainbow. It is already known that when attempting to slow or stop optical signals in nanoscale structures, the key issue to be addressed is how dissipation losses[54,55] and backscattering channels[56,57] affect the light-deceleration ability of these devices. In the waveguide composed of TiO$_2$@PMMA particles, there are dissipation losses and backward scattering, but it is difficult to form back reflection, so the rainbow effect cannot be achieved. "Rainbow" formation in the waveguide composed of Ag/AgCl/TiO$_2$@PMMA particles is precisely due to the back reflection caused by the anomalous GH effect. Figure 5 reacts to the back reflection phenomenon in metamaterial waveguides and the complete back reflection that occurs when reaching the critical opening state. When the incoming port reaches a critical value, the outgoing energy flow is experimentally observed to be zero, indicating a light-stopping state, and the back reflection can be maintained for a long time under continuous energy input. We call the behavior of this state perfect back reflection: an optical black hole, a state when the outgoing light is fully captured. The minimum measurable value of the power meter we used is 1 pW, and the measurements are reliable. The incident energy is stable in the experiment, maintaining the measurement for a long time, the rainbow remains constant, and the outgoing energy is zero, indicating that the light is trapped in the waveguide region. As a comparative experiment, the conclusions were repeatedly calibrated and the energy outflow was determined to be constant to zero over a considerable period of time.

In Fig. 5a, it is obvious that there is a big difference in the normalized optical power, for the TiO$_2$@PMMA waveguide and the green light metamaterial waveguide. When the opening size is over 32 μm, the power density of the TiO$_2$@PMMA waveguide is about 0.8 and that of the green light waveguide is about 0.4, where the energy reduction is exactly caused by propagation loss, that is, the loss of the green light waveguide is much larger than that of TiO$_2$@PMMA waveguide. However, as the opening continues to decrease, light of different incident wavelengths exhibits completely different morphologies. For the measured wavelengths from 473 nm to 671 nm, the energy density decreases; until the opening size of 20 μm, where the energy density of the 589 nm wavelength decreases to 0.2. However, that of 532 nm wavelength behaves differently, and as the opening decreases, the

**Table 1 | Output optical power of green light metamaterial waveguide**

| $d1(\mu m)$ | 473 nm(pW) | 532 nm(pW) | 589 nm(pW) | 632.8 nm(pW) | 671 nm(pW) |
|---|---|---|---|---|---|
| 35.625 | 12 | 10 | 10 | 10 | 12 |
| 35 | 12 | 10 | 10 | 10 | 12 |
| 34.375 | 12 | 10 | 10 | 10 | 12 |
| 33.75 | 12 | 10 | 10 | 10 | 12 |
| 33.125 | 12 | 10 | 10 | 10 | 12 |
| 32.5 | 12 | 9 | 10 | 10 | 12 |
| 31.875 | 12 | 9 | 10 | 10 | 12 |
| 31.25 | 12 | 4 | 10 | 10 | 12 |
| 30.625 | 12 | 4 | 10 | 9 | 11 |
| 30 | 11 | 3 | 8 | 9 | 12 |
| 29.375 | 12 | 3 | 8 | 9 | 11 |
| 28.75 | 11 | 2 | 8 | 9 | 12 |
| 28.125 | 11 | 2 | 8 | 9 | 12 |
| 27.5 | 11 | 2 | 7 | 9 | 12 |
| 26.875 | 11 | 2 | 7 | 9 | 11 |
| 26.25 | 11 | 2 | 7 | 9 | 11 |
| 25.625 | 11 | 2 | 7 | 9 | 11 |
| 25 | 10 | 2 | 7 | 8 | 11 |
| 24.375 | 10 | 1 | 7 | 8 | 11 |
| 23.75 | 10 | 1 | 6 | 8 | 11 |
| 23.125 | 10 | 1 | 6 | 7 | 10 |
| 22.5 | 9 | 1 | 6 | 8 | 10 |
| 21.875 | 10 | 0 | 7 | 8 | 11 |
| 21.25 | 10 | 0 | 6 | 7 | 10 |
| 20.625 | 10 | 1 | 6 | 8 | 10 |
| 20 | 10 | 0 | 6 | 8 | 10 |

energy density decreases drastically, and gradually decreases to zero as the opening becomes smaller than 24 μm. Actually, this is a reflection of the fact that the resonant wavelength of the green light metamaterials is close to the input wavelength, and back reflection occurs, which increases the energy density reduction, and finally, there is a complete back reflection, resulting in zero energy outflow. The results of the red light metamaterial waveguide (Fig. 5b) are similar to those of the green light metamaterial waveguide, and for the broadband sample waveguide, Fig. 5c illustrates the phenomenon remarkably well. As the broadband resonance, the outgoing energy of various wavelengths is significantly reduced compared with the single-frequency response. For example, for 30 μm opening size and under the same incident energy intensity, the outgoing power of 589 nm wavelength in the green and red resonance is 0.3, while in the case of broadband resonance is reduced to 0.15, for this wavelength is within the range of broadband resonance. Moreover, the measured light at 532, 589, and 632.8 nm, all experienced zero outgoing power state. It is clear that the slowing down effect of light in the broadband waveguide is enhanced compared to the single frequency waveguide, and the light is imprisoned in a wider frequency range. Figure 5d provides a qualitative illustration of this phenomenon. I. In the TiO$_2$@PMMA waveguide, the incident light undergoes normal GH shift, and produce normal reflection, where part of the energy is consumed in the light-matter interaction and part of the energy exits from the other end of the waveguide. In metamaterial waveguides: II. For non-resonant conditions, similar to that in TiO$_2$@PMMA waveguides, the incident light undergoes only normal GH shifts, the energy of the outgoing light is lower than that of TiO$_2$@PMMA waveguides due to the higher intrinsic loss of the metamaterial. III. When resonance occurs, the incident light undergoes negative GH shift and produces normal and back reflections, and the energy of the outgoing light further decreases. IV. When

the opening size of the waveguide continues to decrease, i.e., when the angle of incidence decreases further, back reflections are enhanced, normal reflections are weakened, and the energy of the outgoing light becomes even lower. V. When the angle of incidence decreases to the critical value, perfect back reflections of the incident light occur, leading to zero-outgoing-power. The physical mechanism of the "rainbow" is backward reflection caused by particles in the waveguide. The displacement caused by the negative GH effect increases the actual distance traveled by the wave in the waveguide, which leads to a decrease in the group velocity of the wave packet, exhibiting the slow wave effect. The state of zero energy outflow, when it occurs, implies that the light is completely captured and stopped at a fixed position. The negative GH shifts measured in Fig. 2 is 300–700 nm, and it can be estimated that the time required for once displacement is about 1–2.33 fs, i.e., the speed of light traveling through the waveguide is slowed down; when the energy flow is zero, that is, light is stopped.

Our experimental results confirm the frequency-selective rainbow phenomenon due to the negative GH effect of waveguide composed of isotropic metamaterial in the visible band; in particular, we obtain amber rainbows of broadband metamaterials generated by back reflection from weakly interacting system and observe optical black holes formed by perfect back reflection: the phenomenon of stopping light. Experiments have shown that the phenomenon of zero energy outflow can occur in negative refraction waveguides under certain conditions. For the waveguide prepared from TiO$_2$@PMMA, it causes a reduction in the outgoing energy, but the phenomenon of zero-energy-outflow will not occur (Fig. 5). Even for the negative refractive medium waveguide, when the wavelength of the incident light is different from the resonant frequency, a large energy loss can be formed, but no zero-energy-outflow occurs as well (Fig. 5). Obviously, when

**Table 2 | Output optical power of red light metamaterial waveguide**

| $d1(\mu m)$ | 473 nm(pW) | 532 nm(pW) | 589 nm(pW) | 632.8 nm(pW) | 671 nm(pW) |
|---|---|---|---|---|---|
| 35.625 | 11 | 10 | 10 | 9 | 10 |
| 35 | 11 | 10 | 10 | 9 | 10 |
| 34.375 | 11 | 10 | 9 | 8 | 10 |
| 33.75 | 11 | 10 | 9 | 8 | 10 |
| 33.125 | 11 | 10 | 9 | 8 | 10 |
| 32.5 | 11 | 10 | 9 | 8 | 10 |
| 31.875 | 10 | 10 | 9 | 4 | 9 |
| 31.25 | 10 | 10 | 8 | 4 | 9 |
| 30.625 | 10 | 9 | 8 | 3 | 9 |
| 30 | 10 | 9 | 8 | 2 | 9 |
| 29.375 | 10 | 9 | 8 | 2 | 9 |
| 28.75 | 9 | 9 | 8 | 2 | 8 |
| 28.125 | 10 | 9 | 7 | 1 | 8 |
| 27.5 | 10 | 9 | 7 | 1 | 8 |
| 26.875 | 9 | 8 | 7 | 1 | 8 |
| 26.25 | 9 | 8 | 7 | 1 | 8 |
| 25.625 | 9 | 8 | 7 | 1 | 7 |
| 25 | 9 | 8 | 6 | 1 | 7 |
| 24.375 | 8 | 8 | 7 | 1 | 6 |
| 23.75 | 8 | 8 | 7 | 0 | 7 |
| 23.125 | 9 | 8 | 6 | 0 | 6 |
| 22.5 | 8 | 7 | 7 | 0 | 6 |
| 21.875 | 8 | 8 | 7 | 0 | 6 |
| 21.25 | 8 | 7 | 7 | 1 | 6 |
| 20.625 | 8 | 7 | 7 | 1 | 6 |
| 20 | 8 | 7 | 7 | 1 | 6 |

resonant occurs in the negative refraction frequency region, the interaction between the light and the medium leads to a great dissipation of energy, due to the back reflection of the incident light. And when the critical conditions are reached, the incident energy is completely dissipated, resulting in a phenomenon where the energy is continuously input in the waveguide for a long period of time, but no energy is outflowed. Therefore, we call this state of zero energy outflow as the formation of an energy black hole, i.e., an "optical black hole". Certainly, this phenomenon is fundamentally different from the optical black hole spoken of in astronomy.

These analyses show that the trap effect associated with group velocity reduction is clearly different from the system's own losses, and that the reduction in energy density is caused by back reflections formed by the resonance state, whereas the zero-energy-outflow is the result of the complete back reflections. Perfect absorption of metamaterials[58] has been widely studied[59], which achieves basically no reflection for electromagnetic waves incident on the surface of metamaterials through the design principle of impedance matching, and at the same time, the refractive index imaginary part of metamaterials is designed as large as possible, so that all the energy of the electromagnetic waves can be absorbed up to 98%. Different from the perfect absorption, this work is to form the back reflection through the anomalous GH effect of the negative refractive material, and to reach the perfect back reflection at the critical state, so that the outgoing energy of the waveguide becomes zero, which exhibits the optical black hole phenomenon. The present scheme simultaneously allows for high in-coupling efficiencies and broadband, room-temperature operation. The underlying physics of slowing light with negatively refracting media in which light experiences a negative electromagnetic environment in the core layer of the waveguide, forming a negative GH effect[1,12], subverts the conventional slow-wave approach based on

resonance or periodic configurations above the diffraction limit, and highlights the interaction of waves with matter.

Recently, the rainbow trapping has been combined with topological photonics to realize topological rainbows[31–33] in both Hermitian and non-Hermitian[60,61] cases, and to develop toward the study of higher-order angular states[62], broadband, and even multiple rainbows[63]. In addition, broadband "fractal" rainbow trapping[64] has been obtained by combining gradient metamaterial-based acoustic waveguides with fractal spectroscopy. Our work challengingly chose the visible metamaterial platform, relying on fabricated 3D isotropic metamaterials, to achieve not only single rainbow trapping in the visible band, but also to discover the amber rainbow phenomenon in a wider bandwidth. In comparison, the claimed multiple rainbows in topological photonics platform are found to be realized only relying on simulation-designed waveguide beam-splitting paths, which are analogous to the case that multiple light paths produce multiple rainbows (many-to-many); our work experimentally observes the amber rainbow in single light path (one-to-many), which is the result of spatially periodic modulation from weak interactions of disordered assembly metamaterials system, providing a paradigm different from that of waveguide beam-splitting paths.

## Discussion

In summary, we observed an amber rainbow ribbon in an axially varying heterogeneous film waveguide composed of broadband omnidirectional optical metamaterial. Unlike previously reported single rainbow, this amber rainbow ribbon (i.e., broadband light trapping system) significantly enhances light-matter interactions, and the perfect back reflection occurring after the angle of incidence reaches a critical value results in an optical black hole that perfectly absorbs energy effluent radiation. Experiments confirm the slowing down of the group velocity

**Table 3 | Output optical power of broadband sample waveguide and TiO$_2$@PMMA waveguide**

| d1(μm) | Broadband sample | | | | TiO$_2$@PMMA |
| --- | --- | --- | --- | --- | --- |
| | 473 nm(pW) | 532 nm(pW) | 589 nm(pW) | 632.8 nm(pW) | 532 nm(pW) |
| 35.625 | 11 | 9 | 10 | 9 | 27 |
| 35 | 11 | 9 | 10 | 9 | 27 |
| 34.375 | 11 | 9 | 10 | 9 | 27 |
| 33.75 | 9 | 9 | 10 | 6 | 27 |
| 33.125 | 7 | 9 | 10 | 5 | 27 |
| 32.5 | 6 | 9 | 8 | 4 | 27 |
| 31.875 | 5 | 3 | 4 | 4 | 26 |
| 31.25 | 4 | 3 | 4 | 3 | 26 |
| 30.625 | 4 | 3 | 3 | 3 | 26 |
| 30 | 4 | 3 | 4 | 3 | 26 |
| 29.375 | 3 | 3 | 4 | 2 | 26 |
| 28.75 | 4 | 2 | 2 | 2 | 26 |
| 28.125 | 4 | 2 | 2 | 2 | 26 |
| 27.5 | 3 | 1 | 2 | 2 | 26 |
| 26.875 | 4 | 1 | 2 | 2 | 26 |
| 26.25 | 4 | 1 | 2 | 1 | 26 |
| 25.625 | 4 | 0 | 2 | 2 | 26 |
| 25 | 4 | 0 | 1 | 1 | 26 |
| 24.375 | 4 | 0 | 0 | 1 | 25 |
| 23.75 | 4 | 0 | 0 | 1 | 25 |
| 23.125 | 4 | 0 | 0 | 1 | 25 |
| 22.5 | 5 | 1 | 0 | 0 | 25 |
| 21.875 | 5 | 1 | 1 | 0 | 25 |
| 21.25 | 6 | 1 | 1 | 0 | 25 |
| 20.625 | 6 | 1 | 0 | 0 | 25 |
| 20 | 5 | 1 | 1 | 0 | 25 |

of light waves caused by the negative GH effect and the formation of light trapping by perfect back reflection, due to the isotropic visible negative refractive medium. This amber rainbow ribbon phenomenon is caused by the frequency selectivity and spatial modulation when broadband metamaterials responding to white-light illumination. It is another powerful example of possible applications of ultralow loss broadband omnidirectional negative refractive index metamaterials. This work overcomes the great difficulty in trapping broadband visible light by successfully solving the long-standing challenge regarding energy leakage. We expect our work to play a crucial role in the creation of future optical energy harvesting and information processing devices such as quantum optical memories and data processors.

## Methods
### Preparation and characterization of meta-clusters
The Ag/AgCl/TiO$_2$@PMMA meta-cluster particles corresponding to red-light and green-light are prepared using the solvothermal synthesis method[42]. In order to solve the problem of the coating of nano-silver layer of ball-thorn-shaped clusters, AgCl is firstly formed by mixing a certain amount of AgNO$_3$ into TiCl$_4$ during the process of preparing the TiO$_2$ rods. After a photoreduction method, AgCl further disintegrates into elemental chlorine and metallic silver. The latter precipitates on the outer surface of the ball-thorn-shaped structure to form the discrete silver distribution about 1 nm. Next, these agglomerated particles are immersed in PMMA and illuminated to form the Ag/AgCl/TiO$_2$@PMMA particles.

Eight kinds of Ag/AgCl/TiO$_2$@PMMA (meta-cluster) nanoparticles were prepared by solvothermal synthesis from bottom to top[46]. The TEM images of the meta-cluster particles that resonate in the red-, yellow-, and green- light spectra reveal a classic kernel (AgCl/TiO$_2$)-

shell (PMMA) structure. The images confirm the presence of PMMA filling between different nanorods, and the thickness of the PMMA shell is nearly 20–30 nm. According to the idea of multi-frequency composite, the meta-cluster particle system in the wide band of visible light was prepared.

The meta-cluster particle set was characterized by the optical micrograph and SEM images, the TEM images, the local high-angle annular dark-field imaging scanning TEM (HAADF-STEM) images, X-ray diffraction (XRD) patterns, Ultraviolet-visible-near infrared (UV-VIS-NIR) absorption spectra[42,46]. The reliability of the materials, including detailed procedures for sample preparation, can be found in the "Methods" section of ref. 42 and the "Appendix A, B, and C" sections of ref. 46, and the repeatability of the time stability and performance measurements have been repeatedly calibrated.

### Preparation of planar samples
First, A petri dish with a diameter of 10 cm is prepared and filled with deionized water (the liquid level is flush with the mouth of the glass). The prepared Ag/AgCl/TiO$_2$@PMMA particles are dispersed in deionized water to obtain a suspension (the volume ratio of particles to deionized water is about 1:5). Then, anhydrous ethanol is added to the suspension in a volume ratio of 1:3, followed by ultrasonic treatment of the suspension for 5 min to make the metamaterial particles evenly dispersed. The ultrasound-treated suspension is slowly injected onto the surface of ultrapure water using a syringe needle. The gradient of surface tension created by the ultrapure water causes the meta-material particles to spread out rapidly on the water surface to from a complete monolayer film. The film is then transferred to a glass substrate (10 mm × 50 mm). After being naturally air-dried, a uniform planar sample on the glass strip is obtained.

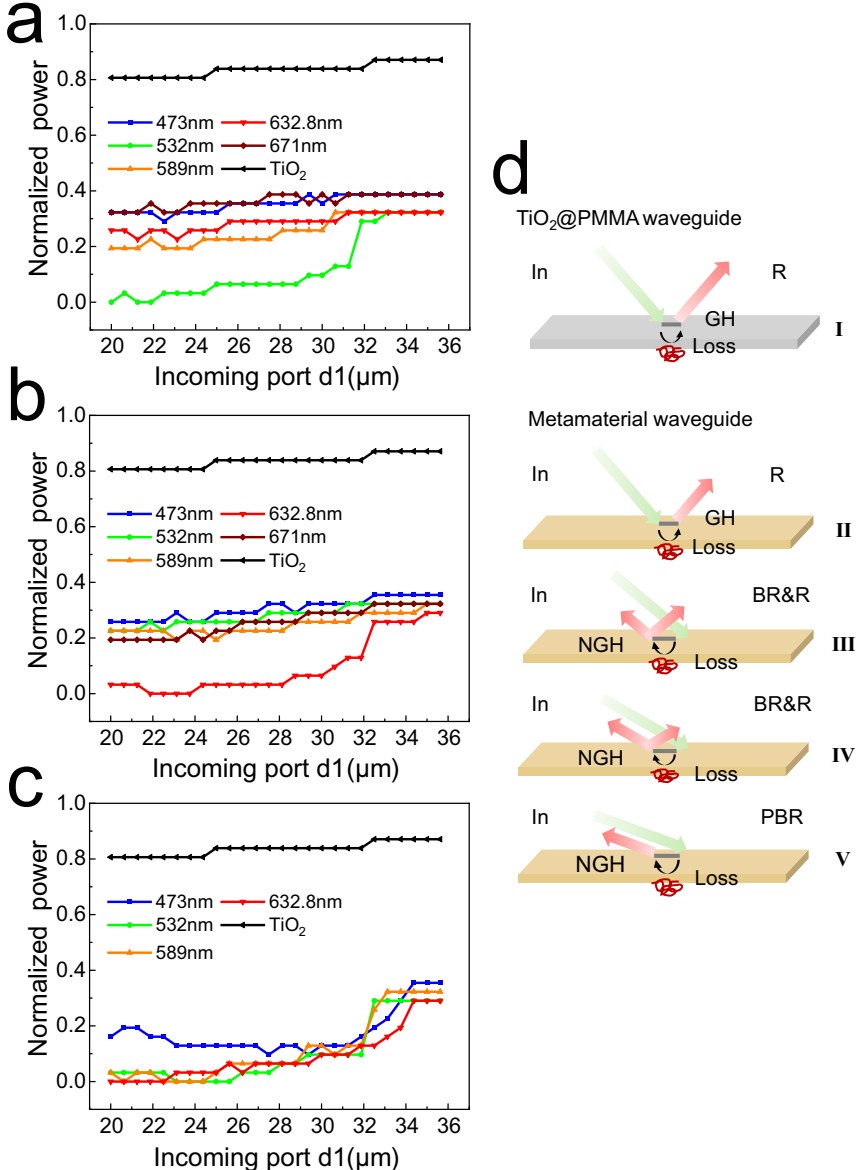

**Fig. 5 | Output optical power measurement. a, b** Normalized output optical power of the green-light metamaterial waveguide and red-light metamaterial waveguide. The output optical power varies with the incoming port $d1$ of the waveguide, and in these two waveguides output optical power of green and red light were observed to decrease to 0, respectively. **c** Normalized output optical power of the broadband sample waveguide. The output optical power of red, yellow, and green light decreases to 0 as the incident port $d1$ varies from 20 to 26 μm. The effective optical power incident into the waveguide was limited to 31 Pw and the measured results are normalized. **d** Schematic diagram of perfect back reflection. In incidence, R reflection, NGH negative GH effect, BR back-reflection, PBR perfect back-reflection.

## Preparation of 3D wedge-shaped samples

A gravity self-assembly device is set as a platform to prepare the wedge-shaped sample. The lifting slab of the experiment platform is adjusted to be horizontal. The 5 mm × 10 mm glass strip is horizontally positioned in the glass substrate, whereas another hydrophilically treated glass strip (20 mm × 40 mm) is vertically placed on the glass strip (5 mm × 10 mm) and pressed down with a proper force to ensure that the suspension will not leak during the painting. Nearly 3.5 μl of the suspension is collected using a pipette and evenly painted from one end to the other along the corner between the two orthogonal glass strips. Under the action of hydrophilicity and gravity, a wedge-shaped suspension is formed. After the water in the wedge-shaped suspension evaporates at room temperature, the horizontal glass strip containing the wedge-shaped sample with Ag/AgCl/TiO$_2$@PMMA particles is taken down.

## Measurement of GH shift

The optical path of measurement of GH shifts is displayed in Fig. 1a. Solid-state lasers (with wavelengths of 473, 532, and 589 nm) and He−Ne lasers (with wavelengths of 632.8 and 671 nm) are used as the light source. GH shifts of the Ag/AgCl/TiO$_2$@PMMA planar samples are measured, and a camera connected to a computer is used to record the interference pattern on the white plate.

## Wedge-shaped optical waveguide consisting of two planar samples

A wedge-shaped waveguide composed of two planar samples is constructed, as demonstrated in the insert of Fig. 3a, and the heights of the incoming and outgoing ports are $d1$ and $d2$ ($d1 > d2$), respectively. Gold foil is used as spacer to control the size of $d2$. The thickness of the wedge-shaped waveguide is adjusted by changing the size of $d1$.

## Data availability

The data supporting the findings of this study are available within the article and are available from the corresponding authors upon request. All data generated in this study are provided in the Source Data file. Source data are provided with this paper.

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

## Acknowledgements

This work was supported by the National Natural Science Foundation of China (Grant Nos. 52272306, 11674267 (X.Z.)).

## Author contributions

J.Z. and X.Z. conceived the idea and designed the model, X.-F.W. and J.Z. performed the simulation study. D.Z., X.X., X.-N.W. and X.-F.W. performed the preparation and characterization of the meta-cluster structure sample, X.Z., X.-F.W., D.Z., X.X. and X.-N.W. performed the optical experiments. X.Z., J.Z. and X.-F.W. drafted the text, aggregated the figures, and wrote the paper with input from all co-authors, X.Z. and J.Z. discussed the results and revised the manuscript.

## Competing interests

The authors declare no competing interests.
