## [Peer Review File · Nature Communications]

Amber rainbow ribbon effect in broadband optical metamaterialsEditorial Note: Parts of this Peer Review File have been redacted as indicated to remove third-party material where no permission to publish could be obtained.

REVIEWER COMMENTS

Reviewer #1 (Remarks to the Author):

This is a nice work, where the authors, building on a series of recent papers by their group, developed a broadband low-loss negative-index metamaterial, which they then used to observe the 'rainbow slowing/stopping' effect in the visible regime.

The authors nicely show the measurement of the negative Goos-Hanchen shift, which is crucial for observing the above phenomenon. This is particularly novel, as I am not aware of any other work that has previously made such a (key) measurement. The authors also report the enhanced power accumulation / harvesting that results from the effect - which, it too, has not been previously reported in similar studies.

Further, the manuscript is well written, and conveys its key points clearly. It is a technically challenging work (particularly because it deals with the visible regime), of broad potential interest, in principle meriting publication in the journal, but only after / if the authors - from my perspective - satisfactorily address the following mandatory points:

1. What is the (negative) refractive-index function vs. frequency? Do its real & imaginary parts obey the Kramers-Kronig relations? It would be helpful to prospective readers if this key result / material property is clearly provided.
2. Was the in-coupled pulse slowed, or stopped? for how long? (i.e., do you also maybe have time-resolved measurements? - that, too, would be helpful to prospective readers)
3. Did you observe back-reflections in the stopped-light regime? Please explain this point clearly.

4. What direct applications could this spatial spectral decomposition have?

5. The discussion of the results of Fig. 5(b) needs to become a bit clearer for the journal's broad readership.

6. The sentence "The underlying physics of the structures upend traditional wave-slowing approaches based on resonances or on periodic configurations above the diffraction limit, while exhibiting greatly boosted density of states and strong wave-matter interactions", on p. 17, needs a more careful justification.

Reviewer #2 (Remarks to the Author):

This paper describes an air waveguide sandwiched by plates coated with disordered particle film which is called metamaterial by the authors. In this study, the thickness of the air core or that of the particle film is tapered and the so-called rainbow effect has been observed. However, I cannot understand this study and argument of this paper. I would list my concerns as follows:

1. One problem is that this study depends on prior studies in so many parts, including metamaterials and their optical characteristics. As this paper cites many papers without detailed explanations, important information cannot be understood at all from this paper. For example, this paper does not show how the used metamaterials work as metamaterials showing negative refraction and how their detailed structures correspond to the reported one. There are neither structural parameters nor theoretical estimations at all in this paper, although the experimental results are sometimes written to be in agreement with theories and the cited papers do not necessarily show such theoretical details.

2. Due to so many citations, the original part of this study is unclear. I wonder if the negative GH shift is said to be new. It is written in the abstract but not in the discussion. As the result is written to agree with Ref. 13, it seems not new. Also, the observation of the trapped rainbow effect might not be new. But this paper spends 2/3 of the total pages for these two. Then, the original part must be the amber rainbow observed for broadband sample. However, this part is very unclear. How the broadband metamaterial contributes the

observation of the repeated rainbow colors is not clearly discussed. The sentence “Obviously, in the region of negative refraction frequency, the interaction between light and medium leads to great dissipation of energy, resulting in the outflow of perfectly absorbed energy to zero and the formation of optical black hole” on lines 300-302 cannot be understood at all.

3. To begin with, I cannot understand the rainbow capture effect. Why such colors are considered as capture. I cannot see any evidence in this paper and prior papers. It seems to me just diffraction or refraction of each wavelength at different position. The authors might consider a situation that the incident wave repeats reflection in-between two plates with angles depending on the wavelength and reaches the emission condition to the vertical direction. Such situation is often said as a slow wave but not like an optical buffer memory. Due to the out-of-plane emission, it is very lossy and cannot be used for in-plane optical devices.

4. Related with 3, why the authors often discuss ultralow loss of this device? There are no evidence of such low loss in this study. In general, slow waves are accompanied by losses. In such a metamaterial with disordered particles, large propagation loss is expected. It should be small when the spacing between plates is as wide as several tens of microns because the number of reflections is limited. Anyway, the argument is too easy without any evidence.

5. In Figs. 4 and 5, the normalized power is presented. I cannot see what power is discussed. It is written as output light power on line 250. How was it output and detected? Was a setup like that depicted in Fig. 3 used? Then, what phenomena are shown in these figures? Besides, the power level is of pW order and very low. Even in a dark room, such lower powers of light in the free space cannot be measured stably without lock-in amplifier.

This paper is overall less informative and the argument lacks evidence and careful discussion that convinces readers.

Reviewer #3 (Remarks to the Author):

This paper reports on an experimental demonstration of an assembly capable of inducing a negative cumulative Goos-Hanchen effect (a spatial shift of a linearly polarized ray undergoing total internal reflection) throughout the visible spectrum.

This setup consists of two films arranged to form a waveguide, with the top plate slightly tilted to create a transverse interaction dependent on the incident wavelength. Particular spherical resonant elements, covered with spikes and studied by the same authors in 2022, seem to be able to exhibit left-handed metamaterial properties, and are arranged on the surface of the waveguide formed to obtain negative Goos-Hanchen shifts. The principle is thus associated with the problem of slow waves, considering the accumulation of propagation delays associated with progression within this structure.

According to the authors' claims, this is:

- The first demonstration of a trapped rainbow effect using a negative-index metamaterial in the visible spectrum.
- The first demonstration of what the authors call an "optical black hole".

This work is being carried out in an incremental context. The proposed set-up is based primarily on the "trapped rainbow" principle. The structure exploited is thus directly linked to the initial principle proposed in 2007 in *Nature*, based on a waveguide made of a negative-index medium whose cross-section gradually decreases to slow down the incident waves. Numerous variants of this set-up were subsequently developed in various fields of electromagnetism and acoustics, a fraction of which are cited in the work of J. Zhao et al.

The elements used to synthesize negative electrical and magnetic properties in a wide bandwidth are based on previous work by the authors.

The effect of introducing partial disorder is highlighted several times, but no reference is given on this point, which seems central to obtaining broadband properties with low dependence on the angle of incidence. It would be useful to highlight the rich and recent literature on the exploitation of disordered hyperuniform media, and to adapt order metrics to your media by direct or indirect methods.

In summary, the central contribution of this paper seems to me to be linked to the exploitation of these particles at the waveguide interface to achieve the trapped rainbow effect, offering a wider bandwidth compared with the exploitation of heterogeneous volume media. The transposition of metamaterial constraints to guide interfaces already has several demonstrations in the literature, but most of the demonstrations I've had a chance to consult are proposed in the microwave domain. The exploitation of a wedge-shaped 3D sample is also reported, but seems more directly linked to previous demonstrations.

I understand the idea of the optical black hole as described by the authors, but the concept does not seem to me to be sufficiently supported by experimental data to be put forward in this way. It seems to me that the extinction effect observed by the authors can be partly explained by the intrinsic losses of the resonators used. How can we clearly separate the trap effect linked to the reduction in group velocity from the system's own losses?

Finally, it seems to me that the information provided does not allow the reproduction of the results presented. It would be useful to propose a more complete description of the methods used to synthesize these meta-structures, as well as a more complete description of the experimental set-ups carried out.

In conclusion, this paper provides an interesting experimental demonstration of the trapped rainbow effect, but I don't think it's possible to recommend its publication in Nature Communications as it stands.

We have responded to the reviewers' comments point by point (in red) in this letter and made the corresponding modifications (in blue) in the revised manuscript. The followings are the responses in detail.

-----Reviewer Comments-----

Reviewer #1 (Remarks to the Author):

This is a nice work, where the authors, building on a series of recent papers by their group, developed a broadband low-loss negative-index metamaterial, which they then used to observe the 'rainbow slowing/stopping' effect in the visible regime.

The authors nicely show the measurement of the negative Goos-Hanchen shift, which is crucial for observing the above phenomenon. This is particularly novel, as I am not aware of any other work that has previously made such a (key) measurement. The authors also report the enhanced power accumulation / harvesting that results from the effect - which, it too, has not been previously reported in similar studies.

Further, the manuscript is well written, and conveys its key points clearly. It is a technically challenging work (particularly because it deals with the visible regime), of broad potential interest, in principle meriting publication in the journal, but only after / if the authors - from my perspective - satisfactorily address the following mandatory points:

1. What is the (negative) refractive-index function vs. frequency? Do its real & imaginary parts obey the Kramers-Kronig relations? It would be helpful to prospective readers if this key result / material property is clearly provided.

Response 1: Thank you very much for your review. In reference 46, we measured the refractive index of a broadband wedge sample in the range of 450–760 nm, and observed negative refractive index in the incident band of 490-730 nm (bandwidth is 240 nm), with the lowest value of -0.41. We have added Fig. 1g, which shows the experimental values and the simulation results of refractive index as a function of frequency. It is the negativity in the real part of the effective refractive index that enables the deceleration of light. We compared the real and imaginary parts of the refractive index of the green and red samples calculated using the simulation model with the results calculated using the KK formula⁴⁹, and it can be seen that they basically satisfy the KK relationship, and the results are shown in the added Fig. 1h. Comparisons were also made using the experimental results, which basically satisfy the KK relationship, but are difficult to plot as curves due to the few data points.

49. Szabo, Z. Closed Form Kramers-Kronig Relations to Extract the Refractive Index of Metamaterials. IEEE Trans. Microwave Theory Tech. 65, 1150-1159 (2017).

Excerpt from Fig. 1 | **g** Refractive index of broadband samples. **h** Simulated refractive index of the green and red metamaterials and approximations of the real part of the refractive index calculated with the Kramers–Kronig integral.

2. Was the in-coupled pulse slowed, or stopped? for how long? (i.e., do you also maybe have time-resolved measurements? - that, too, would be helpful to prospective readers).

Response 2: Thank you very much for your review. In order to study the transmission state of light in a heterogeneous waveguide, we experimentally incident a beam of light with known power along the centreline of the waveguide from the left incoming port (with spacing d_1) to the right outgoing port (with spacing d_2), and the state of the light is judged by the output light power detected at the outgoing port at each wavelength. The light was stopped at the critical thickness as judged from the experimental results where output light power was reduced to zero. The physical mechanism of the “rainbow” is backward reflection caused by particles in the waveguide. The displacement caused by the negative GH effect increases the actual distance travelled by the wave in the waveguide, thus exhibiting a slow wave effect. The state of zero output light power, when it occurs, implies that the light is completely captured and stopped at a fixed position. The negative GH shifts measured in Fig. 2 is 300–700 nm, and it can be estimated that the time required for once displacement is about 1–2.33 fs, i.e., the time required to pass through the waveguide is increased, which corresponds to the slowing down of the group velocity of the light wave; when the energy flow is zero, that is, light is stopped. Since our lasers are not capable of high temporal resolution at different frequencies (theoretical estimates are on the order of picoseconds to femtoseconds), the measurements are done by continuous light sources. The modified Fig. 5 (including the added Fig. 5d) presents in detail the states of slowing down and stopping of light and its principle illustration. In subsequent experiments, we will improve the experimental setup in order to enable direct determination of the temporal resolution of pulses at different wavelengths.

The following sentences have been added to the third paragraph of the “**Perfect back reflection: an optical black hole**” section in the revised manuscript: “The physical mechanism of the “rainbow” is backward reflection caused by particles in the waveguide. The displacement caused by the negative GH effect increases the actual

distance travelled by the wave in the waveguide, which leads to a decrease in the group velocity of the wave packet, exhibiting the slow wave effect. The state of zero energy outflow, when it occurs, implies that the light is completely captured and stopped at a fixed position. The negative GH shifts measured in Fig. 2 is 300–700 nm, and it can be estimated that the time required for once displacement is about 1–2.33 fs, i.e., the speed of light travelling through the waveguide is slowed down; when the energy flow is zero, that is, light is stopped.”

3. Did you observe back-reflections in the stopped-light regime? Please explain this point clearly.

Response 3: Thank you for your review. Fig. 4a–c can illustrate the back-reflection: there is rainbow effect in the thin film waveguide composed of Ag/AgCl/TiO₂@PMMA particles (yellow part), which is the result of back-reflections, while there is no back-reflection in the waveguide composed of TiO₂@PMMA particles (grey part), and therefore it cannot cause the rainbow. It is already known that when attempting to slow or stop optical signals in nanoscale structures, the key issue to be addressed is how dissipation losses^{54,55} and backscattering channels^{56,57} affect the light-deceleration ability of these devices. In the waveguide composed of TiO₂@PMMA particles, there are dissipation losses and backward scattering, but it is difficult to form back reflection, so the rainbow effect cannot be achieved. “Rainbow” formation in the waveguide composed of Ag/AgCl/TiO₂@PMMA particles is precisely due to the back reflection caused by the anomalous GH effect. When the incoming port reaches a critical value, the outgoing energy flow is experimentally observed to be zero, indicating a light-stopping state, and the back reflection can be maintained for a long time under continuous energy input.

The following sentences have been added to the second paragraph of the “**Perfect back reflection: an optical black hole**” section in the revised manuscript: “The back reflection effect in the light-stopping state was observed experimentally. Fig. 4a–c can illustrate that there is a rainbow effect in the thin film waveguide composed of Ag/AgCl/TiO₂@PMMA particles (yellow part), as a result of backward reflection due to the anomalous GH effect, while there is no anomalous GH effect and back reflection occurring in the waveguide composed of TiO₂@PMMA particles (grey part), and therefore it cannot cause the rainbow. It is already known that when attempting to slow or stop optical signals in nanoscale structures, the key issue to be addressed is how dissipation losses^{54, 55} and backscattering channels^{56, 57} affect the light-deceleration ability of these devices. In the waveguide composed of TiO₂@PMMA particles, there are dissipation losses and backward scattering, but it is difficult to form back reflection, so the rainbow effect cannot be achieved. “Rainbow” formation in the waveguide composed of Ag/AgCl/TiO₂@PMMA particles is precisely due to the back reflection caused by the anomalous GH effect. Fig. 5 reacts to the back reflection phenomenon in metamaterial waveguides and the complete back reflection that occurs when reaching the critical opening state. When the incoming port reaches a critical value, the outgoing energy flow is experimentally observed to be zero, indicating a light-stopping state, and the back reflection can be maintained for a long time under continuous energy input.”

54. Tsakmakidis, K. L., Pickering, T. W., Hamm, J. M., Page, A. F., Hess, O. Completely Stopped and Dispersionless Light in Plasmonic Waveguides. *Phys. Rev. Lett.* **112**, 167401 (2014).
55. Archambault, A., Besbes, M., Greffet, J.-J. Superlens in the Time Domain. *Phys. Rev. Lett.* **109**, 097405 (2012).
56. He, S., He, Y., Jin, Y. Revealing the truth about 'trapped rainbow' storage of light in metamaterials. *Sci. Rep.* **2**, 583 (2012).
57. Shen, L., Zheng, X., Deng, X. Stopping terahertz radiation without backscattering over a broad band. *Opt. Express* **23**, 11790-11798 (2015).

Excerpt from Fig. 4| Trapped rainbow effect of 3D wedge-shaped optical waveguide consisting of two planar samples. **a** Schematic of a wedge-shaped metamaterial waveguide composed of two planar samples. Left panel: a 10 mm × 10 mm film composed of Ag/AgCl/TiO₂@PMMA particles (yellow part) and a 10 mm × 10 mm film composed of TiO₂@PMMA particles (gray part) are spin-coated on the left and right sides in the middle of the glass substrate, respectively. Right panel: a 10 mm × 10 mm film composed of Ag/AgCl/TiO₂@PMMA particles (yellow part) is spin-coated on the middle of the glass substrate. Hydrophilically treated glass of 50 mm × 10 mm are used as a substrate. The height of the incoming and outgoing ports is d₁ and d₂ (d₁ > d₂), respectively. The medium between the two planar samples is air. **b** Photos of the trapped rainbow inside the green-light metamaterial waveguide with d₁ = 138 μm (top) and d₁ = 230 μm (bottom). **c** Photos of the trapped rainbow inside the red-light metamaterial waveguide with d₁ = 138 μm (top) and d₁ = 230 μm (bottom).

4. What direct applications could this spatial spectral decomposition have?

Response 4: Thank you for your review. This spatial spectral separation is completely different from the prism dispersion model, and it is an intuitive, pictorial view of the “trapped rainbow” caused by the anomalous GH effect of metamaterial media, which is a direct proof of the theory that light with different frequencies is trapped at different locations in axially varying inhomogeneous waveguides composed of positive-negative refractive-index media. According to the theoretical model proposed by Hess et al.¹² and verified by many subsequent related works¹³⁻²², this trapped rainbow phenomenon opened a new way to slow down and even fix light over a wide frequency range. It offers important potential direct applications in the construction of nanophotonic wavelength routers, optical storage and optical buffer devices.

The following sentences have been added to the last paragraph of the “(b) 3D wedge-shaped optical waveguide consisting of two planar samples” section in the revised

manuscript: “This spatial spectral separation is completely different from the prism dispersion model, and it is an intuitive, pictorial view of the “trapped rainbow” caused by the anomalous GH effect of metamaterial media, which is a direct proof of the theory that light with different frequencies is trapped at different locations in axially varying inhomogeneous waveguides composed of positive-negative refractive-index media. According to the theoretical model proposed by Hess et al.¹² and verified by many subsequent related works¹³⁻²², this trapped rainbow phenomenon opened a new way to slow down and even fix light over a wide frequency range. It offers important potential direct applications in the construction of nanophotonic wavelength routers, optical storage and optical buffer devices.”

5. The discussion of the results of Fig. 5(b) needs to become a bit clearer for the journal's broad readership.

Response 4: Thank you for your review. Regarding the loss of the slow-wave structure, we show quantitative results and analysis in Fig. 5. As can be seen, the loss in the metamaterial (Ag/AgCl/TiO₂@PMMA) waveguide is significantly larger than that in the TiO₂@PMMA waveguide, and there is even a zero-outgoing-energy state at the resonance frequency of metamaterial. However, these phenomena are not due to propagation losses, but back reflection. We integrate the original Figs. 4f, g with Fig. 5b to form a new Fig. 5. To illustrate this phenomenon, the schematic in Fig. 5d is added.

Fig. 5 | Output power measurement. **a, b** Normalized curves of output light power measured at the outgoing port of the green-light metamaterial waveguide and red-light metamaterial waveguide. The output light power varies with the incoming port d_1 of the waveguide, and in these two waveguides output light power of green and red light were observed to decrease to 0, respectively. **c** Normalized output light power of the broadband sample waveguide. When the light power incident into the waveguide is limited to 31 pW, the output light power of red, yellow, and green light decreases to 0 as the incident port d_1 varies from 20 to 26 μm . **d** Schematic diagram of perfect back reflection. In: Incidence, R: reflection, NGH: Negative GH effect, BR: Back reflection, PBR: Perfect back reflection

We have rewritten the narrative of Fig. 5 in the “Perfect back reflection: an optical black hole” section in the revised manuscript: “The 3D wedge-shaped optical waveguides consisting of two planar samples mentioned above are also used for light power measurement, respectively, and the heights of the incoming and outgoing ports are d_1

and d_2 ($d_1 > d_2$, and d_2 is set as $2.4 \mu\text{m}$ here). We used the setup shown in Fig. 3a and measured the variation of the output optical power from the rear port of the waveguide with the size of the waveguide incoming port d_1 , at a known and constant incident power. The experiments were conducted in an optical darkroom, the effective light power incident into the waveguide was limited to 31 pW and a NOVAVI handheld optical power tester with measurement accuracy of 1 pW was used. Tables 1, 2 and 3 show the test data for the output energy flow of the green, red and broadband materials as well as the $\text{TiO}_2\text{@PMMA}$ waveguide, respectively, which were then normalized by taking the incident light power as the maximum value (31 pW) to obtain Fig. 5a-c. We measured the variation of the outgoing optical power with the size of incoming port d_1 of the waveguide, and the results show the state of the light in the waveguide. It can be found that for a fixed outlet $d_2 = 2.4 \mu\text{m}$, the decreasing incident opening of the waveguide leads to variations in the outgoing optical power, and the light is completely stopped when reaching the critical thickness, and the emitted optical power decreases to 0. When the opening size is increased over a certain thickness, the trapped light escapes rapidly, and the outgoing light power increases and tends to stabilize, and the waveguide is almost without the ability of the loss of light, at which time the loss of light power is due to the inherent loss of the waveguide material (if there is no loss of light in the waveguide, the normalized output optical power should be 1).

The back reflection effect in the light-stopping state was observed experimentally. Fig. 4a–c can illustrate that there is a rainbow effect in the thin film waveguide composed of $\text{Ag/AgCl/TiO}_2\text{@PMMA}$ particles (yellow part), as a result of backward reflection due to the anomalous GH effect, while there is no anomalous GH effect and back reflection occurring in the waveguide composed of $\text{TiO}_2\text{@PMMA}$ particles (grey part), and therefore it cannot cause the rainbow. It is already known that when attempting to slow or stop optical signals in nanoscale structures, the key issue to be addressed is how dissipation losses^{54,55} and backscattering channels^{56,57} affect the light-deceleration ability of these devices. In the waveguide composed of $\text{TiO}_2\text{@PMMA}$ particles, there are dissipation losses and backward scattering, but it is difficult to form back reflection, so the rainbow effect cannot be achieved. “Rainbow” formation in the waveguide composed of $\text{Ag/AgCl/TiO}_2\text{@PMMA}$ particles is precisely due to the back reflection caused by the anomalous GH effect. Fig. 5 reacts to the back reflection phenomenon in metamaterial waveguides and the complete back reflection that occurs when reaching the critical opening state. When the incoming port reaches a critical value, the outgoing energy flow is experimentally observed to be zero, indicating a light-stopping state, and the back reflection can be maintained for a long time under continuous energy input. We call the behavior of this state perfect back reflection: an optical black hole, a state when the outgoing light is fully captured. The minimum measurable value of the power meter we used is 1 pW , and the measurements are reliable. The incident energy is stable in the experiment, maintaining the measurement for a long time, the rainbow remains constant, and the outgoing energy is zero, indicating that the light is trapped in the waveguide region. As a comparative experiment, the conclusions were repeatedly calibrated and the energy outflow was determined to be constant to zero over a considerable period of time.

In Fig. 5a, it is obvious that there is a big difference in the normalized optical power, for the TiO₂@PMMA waveguide and the green light metamaterial waveguide. When the opening size is over 32 μm, the power density of the TiO₂@PMMA waveguide is about 0.8 and that of the green light waveguide is about 0.4, where the energy reduction is exactly caused by propagation loss, that is, the loss of the green light waveguide is much larger than that of TiO₂@PMMA waveguide. However, as the opening continues to decrease, light of different incident wavelengths exhibits completely different morphologies. For the measured wavelengths from 473 nm to 671 nm, the energy density decreases; until the opening size of 20 μm, where the energy density of the 589 nm wavelength decreases to 0.2. However, that of 532 nm wavelength behaves differently, and as the opening decreases, the energy density decreases drastically, and gradually decreases to zero as the opening becomes smaller than 24 μm. Actually, this is a reflection of the fact that the resonant wavelength of the green light metamaterials is close to the input wavelength, and back reflection occurs, which increases the energy density reduction, and finally, there is a complete back reflection, resulting in zero energy outflow. The results of the red light metamaterial waveguide (Fig. 5b) are similar to those of the green light metamaterial waveguide, and for the broadband sample waveguide, Fig.5c illustrates the phenomenon remarkably well. As the broadband resonance, the outgoing energy of various wavelengths is significantly reduced compared with the single-frequency response. For example, for 30 μm opening size and under the same incident energy intensity, the outgoing power of 589 nm wavelength in the green and red resonance is 0.3, while in the case of broadband resonance is reduced to 0.15, for this wavelength is within the range of broadband resonance. Moreover, the measured light at 532 nm, 589 nm, and 632.8 nm, all experienced zero outgoing power state. It is clear that the slowing down effect of light in the broadband waveguide is enhanced compared to the single frequency waveguide, and the light is imprisoned in a wider frequency range. Fig. 5d provides a qualitative illustration of this phenomenon. I. In the TiO₂@PMMA waveguide, the incident light undergoes normal GH shift, and produce normal reflection, where part of the energy is consumed in the light-matter interaction and part of the energy exits from the other end of the waveguide. In metamaterial waveguides: II. For non-resonant conditions, similar to that in TiO₂@PMMA waveguides, the incident light undergoes only normal GH shifts, the energy of the outgoing light is lower than that of TiO₂@PMMA waveguides due to the higher intrinsic loss of the metamaterial. III. When resonance occurs, the incident light undergoes negative GH shift and produces normal and back reflections, and the energy of the outgoing light further decreases. IV. When the opening size of the waveguide continues to decrease, i.e., when the angle of incidence decreases further, back reflections are enhanced, normal reflections are weakened, and the energy of the outgoing light becomes even lower. V. When the angle of incidence decreases to the critical value, perfect back reflections of the incident light occur, leading to zero-outgoing-power. The physical mechanism of the “rainbow” is backward reflection caused by particles in the waveguide. The displacement caused by the negative GH effect increases the actual distance travelled by the wave in the waveguide, which leads to a decrease in the group velocity of the wave packet,

exhibiting the slow wave effect. The state of zero energy outflow, when it occurs, implies that the light is completely captured and stopped at a fixed position. The negative GH shifts measured in Fig. 2 is 300–700 nm, and it can be estimated that the time required for once displacement is about 1–2.33 fs, i.e., the speed of light travelling through the waveguide is slowed down; when the energy flow is zero, that is, light is stopped.

Experiments have shown that the phenomenon of zero energy outflow can occur in negative refraction waveguides under certain conditions. For the waveguide prepared from TiO₂@PMMA, it causes a reduction in the outgoing energy, but the phenomenon of zero-energy-outflow will not occur (Fig. 5). Even for the negative refractive medium waveguide, when the wavelength of the incident light is different from the resonant frequency, a large energy loss can be formed, but no zero-energy-outflow occurs as well (Fig. 5). Obviously, when resonant occurs in the negative refraction frequency region, the interaction between the light and the medium leads to a great dissipation of energy, due to the back reflection of the incident light. And when the critical conditions are reached, the incident energy is completely dissipated, resulting in a phenomenon where the energy is continuously input in the waveguide for a long period of time, but no energy is outflowed. Therefore, we call this state of zero energy outflow as the formation of an energy black hole, i.e., an "optical black hole". Certainly, this phenomenon is fundamentally different from the optical black hole spoken of in astronomy.

These analyses show that the trap effect associated with group velocity reduction is clearly different from the system's own losses, and that the reduction in energy density is caused by back reflections formed by the resonance state, whereas the zero-energy-outflow is the result of the complete back reflections. Perfect absorption of metamaterials⁵⁸ has been widely studied⁵⁹, which achieves basically no reflection for electromagnetic waves incident on the surface of metamaterials through the design principle of impedance matching, and at the same time, the refractive index imaginary part of metamaterials is designed as large as possible, so that all the energy of the electromagnetic waves can be absorbed up to 98%. Different from the perfect absorption, this work is to form the back reflection through the anomalous GH effect of the negative refractive material, and to reach the perfect back reflection at the critical state, so that the outgoing energy of the waveguide becomes zero, which exhibits the optical black hole phenomenon. The present scheme simultaneously allows for high in-coupling efficiencies and broadband, room-temperature operation. The underlying physics of slowing light with negatively refracting media in which light experiences a negative electromagnetic environment in the core layer of the waveguide, forming a negative GH effect^{1, 12}, subverts the conventional slow-wave approach based on resonance or periodic configurations above the diffraction limit, and highlights the interaction of waves with matter.”

58. Landy, N. I., Sajuyigbe, S., Mock, J. J., Smith, D. R., Padilla, W. J. Perfect metamaterial absorber. *Phys. Rev. Lett.* **100**, 207402 (2008).

59. Liu, X., Xia, F., Wang, M., Liang, J., Yun, M. Working Mechanism and Progress of

6. The sentence "The underlying physics of the structures upend traditional wave-slowing approaches based on resonances or on periodic configurations above the diffraction limit, while exhibiting greatly boosted density of states and strong wave-matter interactions", on p. 17, needs a more careful justification.

Response 6: Thank you for your suggestion. In our waveguide model, the light is forced by the negative electromagnetic "environment" that it experiences in the core layer of the waveguide and takes a forward "step" in each half period, followed by a negative shift step. At the zero-group velocity point, the negative GH shifts bring the light ray precisely back to its initial position. We originally meant that the realization of slow light with a negatively refracting medium is significantly different from the traditional slow-wave approach based on resonance or periodic configurations above the diffraction limit, and that our approach highlights the interaction of waves with matter.

We have modified the original sentence in the revised manuscript as follows: "The underlying physics of slowing light with negatively refracting media in which light experiences a negative electromagnetic environment in the core layer of the waveguide, forming a negative GH effect^{1, 12}, subverts the conventional slow-wave approach based on resonance or periodic configurations above the diffraction limit, and highlights the interaction of waves with matter."

Reviewer #2 (Remarks to the Author):

This paper describes an air waveguide sandwiched by plates coated with disordered particle film which is called metamaterial by the authors. In this study, the thickness of the air core or that of the particle film is tapered and the so-called rainbow effect has been observed. However, I cannot understand this study and argument of this paper. I would list my concerns as follows:

1. One problem is that this study depends on prior studies in so many parts, including metamaterials and their optical characteristics. As this paper cites many papers without detailed explanations, important information cannot be understood at all from this paper. For example, this paper does not show how the used metamaterials work as metamaterials showing negative refraction and how their detailed structures correspond to the reported one. There are neither structural parameters nor theoretical estimations at all in this paper, although the experimental results are sometimes written to be in agreement with theories and the cited papers do not necessarily show such theoretical details.

Response1: Thank you for your suggestion. We have added information on the structure, morphology, parameters, and optical properties of the metamaterials in Fig. 1, with a critical description in the first paragraph of the **Result** section in the revised manuscript: "Fig. 1a shows the schematic diagram of the construction of axially varying

heterogeneous film waveguides for slowing down light, based on proposed broadband omnidirectional visible metamaterials. Using the monochrome particles red and green⁴² and the combination of eight kinds of particles⁴⁶, monochrome film and broadband film can be prepared respectively (see Methods). The broadband planar film samples are assembled automatically by a list of narrowband, ultra-low loss isotropic meta-cluster system without selection, covering the main bands from green light to red light. The broadband waveguide consists of two planar samples (see Methods).

As has been reported, we have invented an ultralow loss isotropic metamaterial in the visible spectrum⁴², and the ball-thorn-shaped metamaterial cluster model consist of a spherical kernel and many protruding rods (Fig. 1b–c shows green light and red light models). In the simulation, both the kernel and rods are made of TiO₂ coated by Ag of 1 nm in thickness. 600 identical rods with cross-sectional diameter of 15 nm are uniformly distributed around the surface of a kernel. l represents the diameter of the meta-cluster, r is the radius of the spherical kernel, and P refers to the lattice constant of the meta-cluster, the meta-cluster is fully immersed in polymethyl methacrylate (PMMA). The structural parameters of red-light model are $l = 640$ nm, $r = 215$ nm, and $P = 670$ nm, and that of green-light model are $l = 530$ nm, $r = 165$ nm, and $P = 560$ nm. Adjusting l , r , and P according to scaling rule enables the model to be applicable to different working frequency bands. The cluster is named Ag/AgCl/TiO₂@PMMA, and the ball-thorn-shaped meta-clusters with symmetrical structure consisting of the dielectric and its surface dispersed super-thin silver layer have replaced the lithographically defined meta-atoms in existing negative-index metamaterials (NIMs); it is found that the discrete super-thin silver layer produced by the photoreduction method can generate plasmon resonance when excited by electromagnetic waves, thereby achieving the performance of metamaterials. The significant reduction in silver coating thickness provides the physical basis for the decreased joule heating and the realization of ultralow losses. Subsequently, we successfully obtained a novel broadband omnidirectional metamaterial (Fig. 1d shows the unit of broadband omnidirectional meta-clusters system) randomly assembled by a list of narrowband, omnidirectional, and ultralow loss meta-cluster system using a bottom-up approach⁴⁶. Fig. 1e shows the TEM images of the particles that resonate in the green (left) and red (right) light spectrum, revealing a classic kernel–shell structure. The TEM images show that the size of the ball-thorn-shaped Ag/AgCl/TiO₂ particle is approximately 500–700 nm, and the thickness of the PMMA shell is nearly 20–30 nm. The negative refraction for red sample occurs at around 610–640 nm, and the minimum refractive index is -0.41 at 630 nm; the negative refraction for green sample occurs at around 520–550 nm, and the minimum refractive index is about -0.30 at 532 nm. Fig. 1f shows the SEM image of broadband film sample experimentally obtained by disordered self-assembly of eight kinds of single-frequency meta-clusters. For the first time, the negative refractive index of 490 nm–730 nm band (Fig. 1g) and broadband inverse Doppler effect across most of the visible spectrum was observed in the proposed broadband metamaterials. Simulated refractive index of the green and red metamaterials and approximations of the real part of the refractive index calculated with the Kramers–Kronig integral⁵¹ are shown in the Fig. 1h, and they basically satisfy the KK relationship.

In addition, related study¹ has shown that that the negativity in the real part of the effective refractive index enables the deceleration of light in the layers of the heterostructure. On these bases, the research in this paper can be conveniently carried out.”

51. Szabo, Z. Closed Form Kramers-Kronig Relations to Extract the Refractive Index of Metamaterials. *IEEE Trans. Microwave Theory Tech.* **65**, 1150-1159 (2017).

Fig. 1 | Conceptual illustration of the axially varying heterogeneous waveguides based on

proposed broadband omnidirectional visible metamaterials. **a** Schematic diagram of axially varying heterogeneous waveguides composed of broadband planar film samples. **b** Green light meta-cluster model with parameters of $l = 530$ nm, $r = 165$ nm, and $P = 560$ nm (left Figure), the cluster is composed of a spherical core and many prominent rods. When the response band light ray incident, the meta-clusters will occur negative refraction effect. The profile current distribution perpendicular to the external magnetic field and the 1/8 model profile (right picture). **c** Red-light meta-cluster model with parameters of $l = 640$ nm, $r = 215$ nm, and $P = 670$ nm. **d** Cluster unit of broadband omnidirectional meta-clusters system; it is composed of 8 clusters with response bands of 490 nm, 500 nm, 540 nm, 570 nm, 600 nm, 640 nm, 680 nm, and 700 nm, respectively. **e** TEM images of green-light (left) and red-light (right) particles. The size of the ball-thorn-shaped Ag/AgCl/TiO₂ particle is approximately 500–700 nm, and the thickness of the PMMA shell is nearly 20–30 nm. **f** SEM image of the monolayer film of the broadband metamaterial, which is obtained by disordered self-assembly of eight kinds of single-frequency particles. **g** Refractive index curve measured for broadband sample S_B , green-light sample S_G , and red-light sample S_R . S_{Bsim} is the simulated value of the broadband sample. **b**, **c** and **e** cite from Ref. 42; **d**, **f**, and **g** cite from Ref. 46. **h** Simulated refractive index of the green and red metamaterials and approximations of the real part of the refractive index calculated with the Kramers–Kronig integral.

2. Due to so many citations, the original part of this study is unclear. I wonder if the negative GH shift is said to be new. It is written in the abstract but not in the discussion. As the result is written to agree with Ref. 13, it seems not new. Also, the observation of the trapped rainbow effect might not be new. But this paper spends 2/3 of the total pages for these two. Then, the original part must be the amber rainbow observed for broadband sample. However, this part is very unclear. How the broadband metamaterial contributes the observation of the repeated rainbow colors is not clearly discussed. The sentence “Obviously, in the region of negative refraction frequency, the interaction between light and medium leads to great dissipation of energy, resulting in the outflow of perfectly absorbed energy to zero and the formation of optical black hole” on lines 300-302 cannot be understood at all.

2.1. Due to so many citations, the original part of this study is unclear. I wonder if the negative GH shift is said to be new. It is written in the abstract but not in the discussion. As the result is written to agree with Ref. 13, it seems not new.

Response 2.1: Thank you very much for your criticism and suggestions. The original part of this article is the experimental results obtained using negative refractive index metamaterials, including negative GH shifts, trapped rainbow, especially amber rainbows observed in broadband samples and zero energy outflow in waveguides. Many studies have been conducted over the years due to the importance and interest of the GH effect and the trapped rainbow effect. For example, the work in Ref. 14 (incorrectly written as Ref. 13 in the comments), as mentioned by the reviewer, was realized by a metallic dendritic meta-surface. Moreover, past work was only available for materials with narrow bands (as shown in Fig. 2c), i.e., a sample acts only in a specific band. Based on the theory of Ref. 12, the trapped rainbow is generated by the

negative GH effect of metamaterials with negative refractive index. However, the difficulty of obtaining materials with negative refractive index in visible light has constrained the progress of the research, making this study has not been completed in metamaterials. Recently, our group has designed and prepared ball-thorn-shaped ultralow loss, omnidirectional, and broadband visible metamaterials, and experimentally determined the negative refractive index of the three-dimensional materials^{42, 46}, which provides the basis for realizing this work. As pointed out by reviewer 1, “This is a nice work, where the authors, building on a series of recent papers by their group, developed a broadband low-loss negative-index metamaterial, which they then used to observe the 'rainbow slowing/stopping' effect in the visible regime. The authors nicely show the measurement of the negative Goos-Hanchen shift, which is crucial for observing the above phenomenon. This is particularly novel, as I am not aware of any other work that has previously made such a (key) measurement. The authors also report the enhanced power accumulation / harvesting that results from the effect - which, it too, has not been previously reported in similar studies.”

The following paragraph has been added to the second paragraph of the “**Measurement of Goos-Hänchen (GH) shift**” section in the revised manuscript: “Many studies have been conducted over the years due to the importance and interest of the GH effect and the trapped rainbow effect. For example, the work in Ref. 14 was realized by a metallic dendritic meta-surface. Moreover, past work was only available for materials with narrow bands (as shown in Fig. 2c), i.e., a sample acts only in a specific band. Based on the theory of Ref. 12, the trapped rainbow is generated by the negative GH effect of metamaterials with negative refractive index. However, the difficulty of obtaining visible spectrum materials with negative refractive index has constrained the progress of the research, making this study has not been completed in metamaterials. Recently, our group has designed and prepared ball-thorn-shaped ultralow loss, omnidirectional, and broadband visible metamaterials, and experimentally determined the negative refractive index of the three-dimensional materials^{42, 46}, which provides the basis for realizing this work.”

2.2. Also, the observation of the trapped rainbow effect might not be new. But this paper spends 2/3 of the total pages for these two.

Response 2.2: Thank you for your comments. As the behavioral test of the broadband sample is similar to that of the single frequency sample, it leads to the results of the broadband part not being highlighted. Following your suggestion, we have rearranged the layout of the article by integrating the related contents into three parts: GH effect, trapped rainbow and energy flow. In each section, single-frequency and wide-frequency behaviors are introduced separately, and the novel changes and implications arising from broadband sample are carefully compared and discussed. Details are shown in the revised manuscript.

2.3. Then, the original part must be the amber rainbow observed for broadband sample. However, this part is very unclear. How the broadband metamaterial contributes the observation of the repeated rainbow colors is not clearly discussed.

Response 2.3: Thank you for your criticism. The amber rainbow observed in broadband samples is a new phenomenon that has never been reported in the past. We reported this experimental result in the manuscript, but we are sorry that it was not discussed carefully in the previous manuscript. The trapped rainbow effect is based on the anomalous GH effect in negatively refractive media. The rainbow effect occurs in single metamaterials where the particles resonate at the same frequency, such as red and green metamaterials (Fig. 4b, c). For broadband structures consisting of multiple composite metamaterial particles, a rainbow band, i.e. the amber rainbow phenomenon, is formed (Fig. 4f). The physical origin of it is still the anomalous GH effect, where amber rainbow appears due to the difference in resonance wavelengths of the different particles. As in the case of negative refraction appearances, broadband negative refraction results from the weak interactions exhibited by different particles. Here, each particle can produce an anomalous GH effect and appears as a corresponding rainbow, thus overall forming an amber rainbow. These results further illustrate that the behavior of metamaterials is controlled by the constituent structural units. At the same time, the units in the microstructure are randomly arranged, but do not affect the macroscopic overall behavior of the material, which is quite different from strongly interacting systems such as photonic and acoustic crystals. This feature provides considerable convenience for designing metamaterial devices.

The following paragraph has been added to the third paragraph of the “(b) **3D wedge-shaped optical waveguide consisting of two planar samples**” section in the revised manuscript: “The experimental results in Fig. 4 show that the trapped rainbow effect is based on the anomalous GH effect in negatively refractive media. The rainbow effect occurs in single metamaterials where the particles resonate at the same frequency, such as red and green metamaterials (Fig. 4b, c). For broadband structures consisting of multiple composite metamaterial particles, a rainbow band, i.e. the amber rainbow phenomenon, is formed (Fig. 4f). The physical origin of it is still the anomalous GH effect, where amber rainbow appears due to the difference in resonance wavelengths of the different particles. As in the case of negative refraction appearances, broadband negative refraction results from the weak interactions exhibited by different particles. Here, each particle can produce an anomalous GH effect and appears as a corresponding rainbow, thus overall forming an amber rainbow. These results further illustrate that the “rainbow” phenomenon in the system that slows down the group velocity of light is the behavior of the metamaterial, which is controlled by the meta-cluster constituent units. At the same time, the units in the microstructure are randomly arranged, but do not affect the macroscopic overall behavior of the material. This behavior of the weak interaction system is quite different from strongly interacting systems such as photonic and acoustic crystals. This feature provides considerable convenience for designing metamaterial devices.”

2.4. The sentence “Obviously, in the region of negative refraction frequency, the interaction between light and medium leads to great dissipation of energy, resulting in the outflow of perfectly absorbed energy to zero and the formation of optical black hole”

on lines 300-302 cannot be understood at all.

Response 2.4: Experiments have shown that the phenomenon of zero energy outflow can occur in negatively reflecting waveguides under certain conditions. It shows that in the negative reflection frequency region, the interaction between the light and the medium leads to a great dissipation of energy, due to the back reflection of the incident light. And when the critical conditions are reached, the incident energy is completely dissipated, resulting in a phenomenon where the energy is continuously input in the waveguide for a long period of time, but no energy is outflowed. For the waveguide of the same configuration prepared from $\text{TiO}_2\text{@PMMA}$, it causes a reduction in the outgoing energy, but the phenomenon of zero-energy-outflow will not occur (Fig. 5). Even for the negative refractive medium waveguide, when the wavelength of the incident light is different from the resonant frequency, a large energy loss can be formed, but no zero-energy-outflow occurs as well (Fig. 5). Therefore, we call this state of energy outflow as the formation of an energy black hole, i.e., an "optical black hole". Certainly, this phenomenon is fundamentally different from the optical black holes spoken of in astronomy.

The following paragraph has been added to the fourth paragraph of the “**Perfect back reflection: an optical black hole**” section in the revised manuscript: “Experiments have shown that the phenomenon of zero energy outflow can occur in negative refraction waveguides under certain conditions. For the waveguide prepared from $\text{TiO}_2\text{@PMMA}$, it causes a reduction in the outgoing energy, but the phenomenon of zero-energy-outflow will not occur (Fig. 5). Even for the negative refractive medium waveguide, when the wavelength of the incident light is different from the resonant frequency, a large energy loss can be formed, but no zero-energy-outflow occurs as well (Fig. 5). Obviously, when resonant occurs in the negative refraction frequency region, the interaction between the light and the medium leads to a great dissipation of energy, due to the back reflection of the incident light. And when the critical conditions are reached, the incident energy is completely dissipated, resulting in a phenomenon where the energy is continuously input in the waveguide for a long period of time, but no energy is outflowed. Therefore, we call this state of zero energy outflow as the formation of an energy black hole, i.e., an "optical black hole". Certainly, this phenomenon is fundamentally different from the optical black hole spoken of in astronomy.

These analyses show that the trap effect associated with group velocity reduction is clearly different from the system's own losses, and that the reduction in energy density is caused by back reflections formed by the resonance state, whereas the zero-energy-outflow is the result of the complete back reflections. Perfect absorption of metamaterials⁵⁸ has been widely studied⁵⁹, which achieves basically no reflection for electromagnetic waves incident on the surface of metamaterials through the design principle of impedance matching, and at the same time, the refractive index imaginary part of metamaterials is designed as large as possible, so that all the energy of the electromagnetic waves can be absorbed up to 98%. Different from the perfect absorption, this work is to form the back reflection through the anomalous GH effect of the negative refractive material, and to reach the perfect back reflection at the critical

state, so that the outgoing energy of the waveguide becomes zero, which exhibits the optical black hole phenomenon.”

58. Landy, N. I., Sajuyigbe, S., Mock, J. J., Smith, D. R., Padilla, W. J. Perfect metamaterial absorber. *Phys. Rev. Lett.* **100**, 207402 (2008).
59. Liu, X., Xia, F., Wang, M., Liang, J., Yun, M. Working Mechanism and Progress of Electromagnetic Metamaterial Perfect Absorber. *Photonics* **10**, 205 (2023).

3. To begin with, I cannot understand the rainbow capture effect. Why such colors are considered as capture. I cannot see any evidence in this paper and prior papers. It seems to me just diffraction or refraction of each wavelength at different position. The authors might consider a situation that the incident wave repeats reflection in-between two plates with angles depending on the wavelength and reaches the emission condition to the vertical direction. Such situation is often said as a slow wave but not like an optical buffer memory. Due to the out-of-plane emission, it is very lossy and cannot be used for in-plane optical devices.

Response 3: Thank you for your review. As you have pointed out, this phenomenon is often referred to as the slow-wave phenomenon, which exhibits diffraction of each wavelength at a different location. And we have experimentally realized this phenomenon for the first time in the visible band using three-dimensional negative index materials, in particular the broadband rainbow band phenomenon. The theoretical model for realizing slow waves in the infrared band with negative index metamaterials presents the concept of trapped rainbow for the first time (12. 'Trapped rainbow' storage of light in metamaterials. *Nature* 450, 397-401 (2007).), and has been approved by a considerable number of researchers (15. 'Rainbow' Trapping and Releasing at Telecommunication Wavelengths. *Phys. Rev. Lett.* 102, 056801 (2009), 17. Plasmonic Rainbow Trapping Structures for Light Localisation and Spectrum Splitting. *Phys. Rev. Lett.* 107, 207401 (2011).) The expression we have adopted is in line with that of our peers, and in addition, we have obtained experimental results in visible light, shaped like a rainbow in nature, which makes this expression of rainbow seem graphic and apt. In addition, we obtained an experimental result of zero energy flow at the outgoing port, which can indicate that the light is imprisoned, so the term 'trapped' was adopted. As to whether it can be used for in-plane optics, we did not consider it carefully enough, and the statement mentioned in the paper may not be very accurate, which has been deleted.

The following description was added to the fourth paragraph of the “(b) **3D wedge-shaped optical waveguide consisting of two planar samples**” section in the revised manuscript: “The theoretical model for slowing waves with homogenous isotropic negative index metamaterials first presented the concept of trapped rainbow¹² in the infrared band, which exhibits diffraction of each wavelength at a different location, and has been approved by a considerable number of researchers¹³⁻²². This phenomenon in the visible wavelength band has been experimentally achieved at the waveguide interface using our three-dimensional negative index materials⁴², which is more advanced than that reported in the literature for microwave wavelengths. Moreover, the

adoption of a system of weakly interacting particles⁴⁶ at different scales allows the amber rainbow band to be observed for the first time. The experimental phenomenon is shaped like a rainbow in nature, which is more aptly described by the term “trapped rainbow”.”

4. Related with 3, why the authors often discuss ultralow loss of this device? There are no evidence of such low loss in this study. In general, slow waves are accompanied by losses. In such a metamaterial with disordered particles, large propagation loss is expected. It should be small when the spacing between plates is as wide as several tens of microns because the number of reflections is limited. Anyway, the argument is too easy without any evidence.

4.1. Related with 3, why the authors often discuss ultralow loss of this device? There are no evidence of such low loss in this study.

Response 4.1: Thank you for your review. We are sorry that the descriptions "ultra-low loss isotropic meta-cluster" and "ultralow loss broadband omnidirectional negative refractive index metamaterials" mentioned in the manuscript have caused your doubts due to the simple presentation of the material parameters. As known, the preparation of three-dimensional optical metamaterials has long been a bottleneck in the discipline, and the key difficulty lies in the high loss of the metallic structure. In previous studies, i.e., in Ref. 42 and Ref. 46, we prepared a novel three-dimensional metamaterial for the visible spectrum and obtained for the first time high refractive index quality factors by theoretical calculations and experimental measurements, and these results demonstrated the ultra-low-loss properties of meta-clusters⁴² and broadband metamaterials⁴⁶, and obtained experimental measurements of negative refractive index. The “ultra-low loss” mentioned refers to the properties of this metamaterial for the preparation of the waveguide structure, not to the propagation loss of the waveguide structure.

4.2. In general, slow waves are accompanied by losses. In such a metamaterial with disordered particles, large propagation loss is expected. It should be small when the spacing between plates is as wide as several tens of microns because the number of reflections is limited. Anyway, the argument is too easy without any evidence.

Response 4.2: Thank you for your review. Regarding the loss of the slow-wave structure, we show quantitative results and analysis in Fig. 5. As can be seen, the loss in the metamaterial (Ag/AgCl/TiO₂@PMMA) waveguide is significantly larger than that in the TiO₂@PMMA waveguide, and there is even a zero-outgoing-energy state at the resonance frequency of metamaterial. However, these phenomena are not due to propagation losses, but back reflection. We integrate the original Figs. 4f, g with Fig. 5b to form a new Fig. 5. To illustrate this phenomenon, the schematic in Fig. 5d is added. Fig. 5 reacts to the back reflection phenomenon in metamaterial waveguides and the complete back reflection that occurs when reaching the critical opening state. We call

the behavior of this state perfect back reflection: an optical black hole, a state when the outgoing light is fully captured. The minimum measurable value of the power meter we used is 1 pW, and the measurements are reliable. The incident energy is stable in the experiment, maintaining the measurement for a long time, the rainbow remains constant, and the outgoing energy is zero, indicating that the light is trapped in the waveguide region. As a comparative experiment, the conclusions were repeatedly calibrated and the energy outflow was determined to be constant to zero over a considerable period of time.

In Fig. 5a, it is obvious that there is a big difference in the normalized optical power, for the TiO₂@PMMA waveguide and the green light metamaterial waveguide. When the opening size is over 32 μm, the power density of the TiO₂@PMMA waveguide is about 0.8 and that of the green light waveguide is about 0.4, where the energy reduction is exactly caused by propagation loss, that is, the loss of the green light waveguide is much larger than that of TiO₂@PMMA waveguide. However, as the opening continues to decrease, light of different incident wavelengths exhibits completely different morphologies. For the measured wavelengths from 473 nm to 671 nm, the energy density decreases; until the opening size of 20 μm, where the energy density of the 589 nm wavelength decreases to 0.2. However, that of 532 nm wavelength behaves differently, and as the opening decreases, the energy density decreases drastically, and gradually decreases to zero as the opening becomes smaller than 24 μm. Actually, this is a manifestation of the fact that the resonant wavelength of the green light metamaterials is close to the input wavelength, and back reflection occurs, which increases the energy density reduction, and finally, there is a complete back reflection, resulting in zero energy outflow. The results of the red light metamaterial waveguide (Fig. 5b) are similar to those of the green light metamaterial waveguide, and for the broadband sample waveguide, Fig.5c illustrates the phenomenon remarkably well. As the broadband resonance, the outgoing energy of various wavelengths is significantly reduced compared with the single-frequency response. For example, for 30 μm opening size and under the same incident energy intensity, the outgoing power of 589 nm wavelength in the green and red resonance is 0.3, while in the case of broadband resonance is reduced to 0.15, for this wavelength is within the range of broadband resonance. Moreover, the measured light at 532 nm, 589 nm, and 632 nm, all experienced zero outgoing power state. It is clear that the slowing down effect of light in the broadband waveguide is enhanced compared to the single frequency waveguide, and the light is imprisoned in a wider frequency range.

Fig. 5d provides a qualitative illustration of this phenomenon. I. In the TiO₂@PMMA waveguide, the incident light undergoes normal GH shift, and produce normal reflection, where part of the energy is consumed in the light-matter interaction and part of the energy exits from the other end of the waveguide. In metamaterial waveguides: II. For non-resonant conditions, similar to in TiO₂@PMMA waveguides, the incident light undergoes only normal GH shifts, the energy of the outgoing light is lower than that of TiO₂@PMMA waveguides due to the higher intrinsic loss of the metamaterial. III. When resonance occurs, the incident light undergoes negative GH shift and produces normal and back reflections, and the energy of the outgoing light

further decreases. IV. When the opening size of the waveguide continues to decrease, i.e., when the angle of incidence decreases further, back reflections are enhanced, normal reflections are weakened, and the energy of the outgoing light becomes even lower. V. When the angle of incidence decreases to the critical value, perfect back reflections of the incident light occur, leading to zero-outgoing-power. These experimental results show that the trap effect associated with group velocity reduction is clearly different from the system's own losses, and that the reduction in energy density is caused by back reflections formed by the resonance state, whereas the zero energy outflow is the result of the complete back reflections. Perfect absorption of metamaterials⁵⁸ has been widely studied⁵⁹, which achieves basically no reflection for electromagnetic waves incident on the surface of metamaterials through the design principle of impedance matching, and at the same time, the refractive index imaginary part of metamaterials is designed as large as possible, so that all the energy of the electromagnetic waves can be absorbed up to 98%. Different from the perfect absorption, this work is to form the back reflection through the anomalous GH effect of the negative refractive material, and to reach the perfect back reflection at the critical state, so that the outgoing energy of the waveguide becomes zero, which exhibits the optical black hole phenomenon.

Fig. 5 | Output power measurement. **a, b** Normalized curves of output light power measured at the outgoing port of the green-light metamaterial waveguide and red-light metamaterial waveguide. The output light power varies with the incoming port d_1 of the waveguide, and in these two waveguides output light power of green and red light were observed to decrease to 0, respectively. **c** Normalized output light power of the broadband sample waveguide. When the light power incident into the waveguide is limited to 31 pW, the output light power of red, yellow, and green light decreases to 0 as the incident port d_1 varies from 20 to 26 μm . **d** Schematic diagram of perfect back reflection. In: Incidence, R: reflection, NGH: Negative GH effect, BR: Back reflection, PBR: Perfect back reflection.

5. In Figs. 4 and 5, the normalized power is presented. I cannot see what power is discussed. It is written as output light power on line 250. How was it output and detected? Was a setup like that depicted in Fig. 3 used? Then, what phenomena are shown in these

figures? Besides, the power level is of pW order and very low. Even in a dark room, such lower powers of light in the free space cannot be measured stably without lock-in amplifier.

This paper is overall less informative and the argument lacks evidence and careful discussion that convinces readers.

5.1. In Figs. 4 and 5, the normalized power is presented. I cannot see what power is discussed. It is written as output light power on line 250. How was it output and detected? Was a setup like that depicted in Fig. 3 used? Then, what phenomena are shown in these figures?

Response 5.1: We used the setup shown in Fig. 3a and measured the variation of the output optical power from the rear port of the waveguide with the size of the waveguide incoming port d1, at a known and constant incident power. Specifically, light incident on an axially varying heterogeneous metamaterial waveguide generates, for example, negative GH effect and trapped rainbow effect, which are related to the thickness of the waveguide (achieved by varying the size of incoming port d1), thus resulting in variations in the light power exiting the rear port of the waveguide, and accordingly, the state of light in the waveguide can be analyzed through the detection of exiting optical power.

We have added experimental measurements to the revised manuscript: Table 1. Output optical power of green light metamaterial waveguide, Table 2. Output optical power of red light metamaterial waveguide, and Table 3. Output optical power of broadband sample waveguide and TiO₂@PMMA waveguide. Then, the measured optical power is normalized by taking the incident light power as the maximum value, and the results are shown in Fig. 5 (Original Figs. 4,5). It can be found that for a fixed outlet d2 = 2.4 μm, the decreasing incident opening of the waveguide leads to variations in the outgoing optical power, and the light is completely stopped when reaching the critical thickness, and the emitted optical power decreases to 0. When the opening size is increased over a certain thickness, the trapped light escapes rapidly, and the outgoing light power increases and tends to stabilise, and the waveguide is almost without the ability of the loss of light, at which time the loss of light power is due to the inherent loss of the waveguide material (if there is no loss of light in the waveguide, the normalized output optical power should be 1). Fig. 5 reacts to the back reflection phenomenon in metamaterial waveguides and the complete back reflection that occurs when reaching the critical opening state. We call the behavior of this state perfect back reflection: an optical black hole, a state when the outgoing light is fully captured.

5.2. Besides, the power level is of pW order and very low. Even in a dark room, such lower powers of light in the free space cannot be measured stably without lock-in amplifier.

Response 5.2: The minimum measurable value of the power meter we used is 1 pW, and the measurements are reliable. We have added experimental measurements to the revised manuscript: Table 1. Output optical power of green light metamaterial

waveguide, Table 2. Output optical power of red light metamaterial waveguide, and Table 3. Output optical power of broadband sample waveguide and TiO₂@PMMA waveguide. The incident energy is stable in the experiment, maintaining the measurement for a long time, the rainbow remains constant, and the outgoing energy is zero, indicating that the light is trapped in the waveguide region. As a comparative experiment, the conclusions were repeatedly calibrated and the energy outflow was determined to be constant to zero over a considerable period of time.

Table 1. Output optical power of green light metamaterial waveguide

d1(μm)	473nm(pW)	532nm(pW)	589nm(pW)	632.8nm(pW)	671nm(pW)
35.625	12	10	10	10	12
35	12	10	10	10	12
34.375	12	10	10	10	12
33.75	12	10	10	10	12
33.125	12	10	10	10	12
32.5	12	9	10	10	12
31.875	12	9	10	10	12
31.25	12	4	10	10	12
30.625	12	4	10	9	11
30	11	3	8	9	12
29.375	12	3	8	9	11
28.75	11	2	8	9	12
28.125	11	2	8	9	12
27.5	11	2	7	9	12
26.875	11	2	7	9	11
26.25	11	2	7	9	11
25.625	11	2	7	9	11
25	10	2	7	8	11
24.375	10	1	7	8	11
23.75	10	1	6	8	11
23.125	10	1	6	7	10
22.5	9	1	6	8	10
21.875	10	0	7	8	11
21.25	10	0	6	7	10
20.625	10	1	6	8	10
20	10	0	6	8	10

Table 2. Output optical power of red light metamaterial waveguide

d1(μm)	473nm(pW)	532nm(pW)	589nm(pW)	632.8nm(pW)	671nm(pW)
35.625	11	10	10	9	10
35	11	10	10	9	10
34.375	11	10	9	8	10
33.75	11	10	9	8	10
33.125	11	10	9	8	10
32.5	11	10	9	8	10
31.875	10	10	9	4	9
31.25	10	10	8	4	9
30.625	10	9	8	3	9
30	10	9	8	2	9
29.375	10	9	8	2	9
28.75	9	9	8	2	8
28.125	10	9	7	1	8
27.5	10	9	7	1	8
26.875	9	8	7	1	8
26.25	9	8	7	1	8
25.625	9	8	7	1	7
25	9	8	6	1	7
24.375	8	8	7	1	6
23.75	8	8	7	0	7
23.125	9	8	6	0	6
22.5	8	7	7	0	6
21.875	8	8	7	0	6
21.25	8	7	7	1	6
20.625	8	7	7	1	6
20	8	7	7	1	6

Table 3. Output optical power of broadband sample waveguide and TiO₂@PMMA waveguide

d1(μm)	Broadband sample				TiO ₂ @PMMA
	473nm(pW)	532nm(pW)	589nm(pW)	632.8nm(pW)	532nm(pW)
35.625	11	9	10	9	27
35	11	9	10	9	27
34.375	11	9	10	9	27
33.75	9	9	10	6	27
33.125	7	9	10	5	27
32.5	6	9	8	4	27
31.875	5	3	4	4	26
31.25	4	3	4	3	26
30.625	4	3	3	3	26
30	4	3	4	3	26
29.375	3	3	4	2	26
28.75	4	2	2	2	26
28.125	4	2	2	2	26
27.5	3	1	2	2	26
26.875	4	1	2	2	26
26.25	4	1	2	1	26
25.625	4	0	2	2	26
25	4	0	1	1	26
24.375	4	0	0	1	25
23.75	4	0	0	1	25
23.125	4	0	0	1	25
22.5	5	1	0	0	25
21.875	5	1	1	0	25
21.25	6	1	1	0	25
20.625	6	1	0	0	25
20	5	1	1	0	25

We have rewritten the “**Perfect back reflection: an optical black hole**” section in the revised manuscript: “The 3D wedge-shaped optical waveguides consisting of two planar samples mentioned above are also used for light power measurement, respectively, and the heights of the incoming and outgoing ports are d1 and d2 (d1>d2, and d2 is set as 2.4 μm here). We used the setup shown in Fig. 3a and measured the variation of the output optical power from the rear port of the waveguide with the size of the waveguide incoming port d1, at a known and constant incident power. The experiments were conducted in an optical darkroom, the effective light power incident into the waveguide was limited to 31 pW and a NOVAVII handheld optical power tester with measurement accuracy of 1 pW was used. Tables 1, 2 and 3 show the test data for the output energy flow of the green, red and broadband materials as well as the TiO₂@PMMA waveguide, respectively, which were then normalized by taking the

incident light power as the maximum value (31 pW) to obtain Fig. 5a-c. We measured the variation of the outgoing optical power with the size of incoming port d_1 of the waveguide, and the results show the state of the light in the waveguide. It can be found that for a fixed outlet $d_2 = 2.4 \mu\text{m}$, the decreasing incident opening of the waveguide leads to variations in the outgoing optical power, and the light is completely stopped when reaching the critical thickness, and the emitted optical power decreases to 0. When the opening size is increased over a certain thickness, the trapped light escapes rapidly, and the outgoing light power increases and tends to stabilize, and the waveguide is almost without the ability of the loss of light, at which time the loss of light power is due to the inherent loss of the waveguide material (if there is no loss of light in the waveguide, the normalized output optical power should be 1).

The back reflection effect in the light-stopping state was observed experimentally. Fig. 4a–c can illustrate that there is a rainbow effect in the thin film waveguide composed of Ag/AgCl/TiO₂@PMMA particles (yellow part), as a result of backward reflection due to the anomalous GH effect, while there is no anomalous GH effect and back reflection occurring in the waveguide composed of TiO₂@PMMA particles (grey part), and therefore it cannot cause the rainbow. It is already known that when attempting to slow or stop optical signals in nanoscale structures, the key issue to be addressed is how dissipation losses^{54,55} and backscattering channels^{56,57} affect the light-deceleration ability of these devices. In the waveguide composed of TiO₂@PMMA particles, there are dissipation losses and backward scattering, but it is difficult to form back reflection, so the rainbow effect cannot be achieved. “Rainbow” formation in the waveguide composed of Ag/AgCl/TiO₂@PMMA particles is precisely due to the back reflection caused by the anomalous GH effect. Fig. 5 reacts to the back reflection phenomenon in metamaterial waveguides and the complete back reflection that occurs when reaching the critical opening state. When the incoming port reaches a critical value, the outgoing energy flow is experimentally observed to be zero, indicating a light-stopping state, and the back reflection can be maintained for a long time under continuous energy input. We call the behavior of this state perfect back reflection: an optical black hole, a state when the outgoing light is fully captured. The minimum measurable value of the power meter we used is 1 pW, and the measurements are reliable. The incident energy is stable in the experiment, maintaining the measurement for a long time, the rainbow remains constant, and the outgoing energy is zero, indicating that the light is trapped in the waveguide region. As a comparative experiment, the conclusions were repeatedly calibrated and the energy outflow was determined to be constant to zero over a considerable period of time.

In Fig. 5a, it is obvious that there is a big difference in the normalized optical power, for the TiO₂@PMMA waveguide and the green light metamaterial waveguide. When the opening size is over $32 \mu\text{m}$, the power density of the TiO₂@PMMA waveguide is about 0.8 and that of the green light waveguide is about 0.4, where the energy reduction is exactly caused by propagation loss, that is, the loss of the green light waveguide is much larger than that of TiO₂@PMMA waveguide. However, as the opening continues to decrease, light of different incident wavelengths exhibits completely different morphologies. For the measured wavelengths from 473 nm to 671

nm, the energy density decreases; until the opening size of 20 μm , where the energy density of the 589 nm wavelength decreases to 0.2. However, that of 532 nm wavelength behaves differently, and as the opening decreases, the energy density decreases drastically, and gradually decreases to zero as the opening becomes smaller than 24 μm . Actually, this is a reflection of the fact that the resonant wavelength of the green light metamaterials is close to the input wavelength, and back reflection occurs, which increases the energy density reduction, and finally, there is a complete back reflection, resulting in zero energy outflow. The results of the red light metamaterial waveguide (Fig. 5b) are similar to those of the green light metamaterial waveguide, and for the broadband sample waveguide, Fig.5c illustrates the phenomenon remarkably well. As the broadband resonance, the outgoing energy of various wavelengths is significantly reduced compared with the single-frequency response. For example, for 30 μm opening size and under the same incident energy intensity, the outgoing power of 589 nm wavelength in the green and red resonance is 0.3, while in the case of broadband resonance is reduced to 0.15, for this wavelength is within the range of broadband resonance. Moreover, the measured light at 532 nm, 589 nm, and 632.8 nm, all experienced zero outgoing power state. It is clear that the slowing down effect of light in the broadband waveguide is enhanced compared to the single frequency waveguide, and the light is imprisoned in a wider frequency range. Fig. 5d provides a qualitative illustration of this phenomenon. I. In the $\text{TiO}_2\text{@PMMA}$ waveguide, the incident light undergoes normal GH shift, and produce normal reflection, where part of the energy is consumed in the light-matter interaction and part of the energy exits from the other end of the waveguide. In metamaterial waveguides: II. For non-resonant conditions, similar to that in $\text{TiO}_2\text{@PMMA}$ waveguides, the incident light undergoes only normal GH shifts, the energy of the outgoing light is lower than that of $\text{TiO}_2\text{@PMMA}$ waveguides due to the higher intrinsic loss of the metamaterial. III. When resonance occurs, the incident light undergoes negative GH shift and produces normal and back reflections, and the energy of the outgoing light further decreases. IV. When the opening size of the waveguide continues to decrease, i.e., when the angle of incidence decreases further, back reflections are enhanced, normal reflections are weakened, and the energy of the outgoing light becomes even lower. V. When the angle of incidence decreases to the critical value, perfect back reflections of the incident light occur, leading to zero-outgoing-power. The physical mechanism of the “rainbow” is backward reflection caused by particles in the waveguide. The displacement caused by the negative GH effect increases the actual distance travelled by the wave in the waveguide, which leads to a decrease in the group velocity of the wave packet, exhibiting the slow wave effect. The state of zero energy outflow, when it occurs, implies that the light is completely captured and stopped at a fixed position. The negative GH shifts measured in Fig. 2 is 300–700 nm, and it can be estimated that the time required for once displacement is about 1–2.33 fs, i.e., the speed of light travelling through the waveguide is slowed down; when the energy flow is zero, that is, light is stopped.

Experiments have shown that the phenomenon of zero energy outflow can occur in negative refraction waveguides under certain conditions. For the waveguide prepared

from TiO₂@PMMA, it causes a reduction in the outgoing energy, but the phenomenon of zero-energy-outflow will not occur (Fig. 5). Even for the negative refractive medium waveguide, when the wavelength of the incident light is different from the resonant frequency, a large energy loss can be formed, but no zero-energy-outflow occurs as well (Fig. 5). Obviously, when resonant occurs in the negative refraction frequency region, the interaction between the light and the medium leads to a great dissipation of energy, due to the back reflection of the incident light. And when the critical conditions are reached, the incident energy is completely dissipated, resulting in a phenomenon where the energy is continuously input in the waveguide for a long period of time, but no energy is outflowed. Therefore, we call this state of zero energy outflow as the formation of an energy black hole, i.e., an "optical black hole". Certainly, this phenomenon is fundamentally different from the optical black hole spoken of in astronomy.

These analyses show that the trap effect associated with group velocity reduction is clearly different from the system's own losses, and that the reduction in energy density is caused by back reflections formed by the resonance state, whereas the zero-energy-outflow is the result of the complete back reflections. Perfect absorption of metamaterials⁵⁸ has been widely studied⁵⁹, which achieves basically no reflection for electromagnetic waves incident on the surface of metamaterials through the design principle of impedance matching, and at the same time, the refractive index imaginary part of metamaterials is designed as large as possible, so that all the energy of the electromagnetic waves can be absorbed up to 98%. Different from the perfect absorption, this work is to form the back reflection through the anomalous GH effect of the negative refractive material, and to reach the perfect back reflection at the critical state, so that the outgoing energy of the waveguide becomes zero, which exhibits the optical black hole phenomenon. The present scheme simultaneously allows for high in-coupling efficiencies and broadband, room-temperature operation. The underlying physics of slowing light with negatively refracting media in which light experiences a negative electromagnetic environment in the core layer of the waveguide, forming a negative GH effect^{1, 12}, subverts the conventional slow-wave approach based on resonance or periodic configurations above the diffraction limit, and highlights the interaction of waves with matter.”

58. Landy, N. I., Sajuyigbe, S., Mock, J. J., Smith, D. R., Padilla, W. J. Perfect metamaterial absorber. *Phys. Rev. Lett.* **100**, 207402 (2008).

59. Liu, X., Xia, F., Wang, M., Liang, J., Yun, M. Working Mechanism and Progress of Electromagnetic Metamaterial Perfect Absorber. *Photonics* **10**, 205 (2023).

Reviewer #3 (Remarks to the Author):

This paper reports on an experimental demonstration of an assembly capable of inducing a negative cumulative Goos Hanchen effect (a spatial shift of a linearly polarized ray undergoing total internal reflection) throughout the visible spectrum.

This setup consists of two films arranged to form a waveguide, with the top plate slightly tilted to create a transverse interaction dependent on the incident wavelength. Particular spherical resonant elements, covered with spikes and studied by the same authors in 2022, seem to be able to exhibit left-handed metamaterial properties, and are arranged on the surface of the waveguide formed to obtain negative Goos-Hanchen shifts. The principle is thus associated with the problem of slow waves, considering the accumulation of propagation delays associated with progression within this structure.

According to the authors' claims, this is:

- The first demonstration of a trapped rainbow effect using a negative-index metamaterial in the visible spectrum.
- The first demonstration of what the authors call an "optical black hole".

This work is being carried out in an incremental context. The proposed set-up is based primarily on the "trapped rainbow" principle. The structure exploited is thus directly linked to the initial principle proposed in 2007 in Nature, based on a waveguide made of a negative-index medium whose cross-section gradually decreases to slow down the incident waves. Numerous variants of this set-up were subsequently developed in various fields of electromagnetism and acoustics, a fraction of which are cited in the work of J. Zhao et al.

The elements used to synthesize negative electrical and magnetic properties in a wide bandwidth are based on previous work by the authors.

1. The effect of introducing partial disorder is highlighted several times, but no reference is given on this point, which seems central to obtaining broadband properties with low dependence on the angle of incidence. It would be useful to highlight the rich and recent literature on the exploitation of disordered hyperuniform media, and to adapt order metrics to your media by direct or indirect methods.

Response 1: Thanks to your suggestion, we have added the recent literature on disordered hyperuniform media, which propose the availability of ordering metrics. We have added a statement in the **introduction** section in the revised manuscript: "Our partial disorder effect actually relies on disordered hyperuniform media of recent years, and related order metrics^{47, 48}, and these provide the core of broadband properties with low dependence on the incidence angle."

47. Torquato, S. Disordered hyperuniform heterogeneous materials. *Journal of Physics-Condensed Matter* **28**, 414012 (2016).

48. Torquato, S. Extraordinary disordered hyperuniform multifunctional composites. *J. Compos. Mater.* **56**, 3635-3649 (2022).

2. In summary, the central contribution of this paper seems to me to be linked to the

exploitation of these particles at the waveguide interface to achieve the trapped rainbow effect, offering a wider bandwidth compared with the exploitation of heterogeneous volume media. The transposition of metamaterial constraints to guide interfaces already has several demonstrations in the literature, but most of the demonstrations I've had a chance to consult are proposed in the microwave domain. The exploitation of a wedge-shaped 3D sample is also reported, but seems more directly linked to previous demonstrations.

2.1. In summary, the central contribution of this paper seems to me to be linked to the exploitation of these particles at the waveguide interface to achieve the trapped rainbow effect, offering a wider bandwidth compared with the exploitation of heterogeneous volume media.

Response 2.1: Thank you for your understanding. In contrast to the trapped rainbow mentioned in the literature, we did realize this phenomenon by these particles, especially the first time to realize observable trapped rainbow in visible frequency, breaking the limitation of microwave band. Moreover, the adoption of the particle system has allowed the observation of a broadband rainbow band for the first time.

2.2. The transposition of metamaterial constraints to guide interfaces already has several demonstrations in the literature, but most of the demonstrations I've had a chance to consult are proposed in the microwave domain.

Response 2.2: Our work is the first broadband experimental result for optical frequencies. We have added a statement in the revised manuscript: “This phenomenon in the visible wavelength band has been experimentally achieved at the waveguide interface using our three-dimensional negative index materials⁴², which is more advanced than that reported in the literature for microwave wavelengths. Moreover, the adoption of a system of weakly interacting particles⁴⁶ at different scales allows the amber rainbow band to be observed for the first time.”

3. I understand the idea of the optical black hole as described by the authors, but the concept does not seem to me to be sufficiently supported by experimental data to be put forward in this way. It seems to me that the extinction effect observed by the authors can be partly explained by the intrinsic losses of the resonators used. How can we clearly separate the trap effect linked to the reduction in group velocity from the system's own losses?

Response 3: Thank you for your review. Regarding the loss of the slow-wave structure, we show quantitative results and analysis in Fig. 5. As can be seen, the loss in the metamaterial (Ag/AgCl/TiO₂@PMMA) waveguide is significantly larger than that in the TiO₂@PMMA waveguide, and there is even a zero-outgoing-energy state at the resonance frequency of metamaterial. However, these phenomena are not due to propagation losses, but back reflection. We integrate the original Figs. 4f, g with Fig. 5b to form a new Fig. 5. To illustrate this phenomenon, the schematic in Fig. 5d is added.

Fig. 5 reacts to the back reflection phenomenon in metamaterial waveguides and the complete back reflection that occurs when reaching the critical opening state. We call the behavior of this state perfect back reflection: an optical black hole, a state when the outgoing light is fully captured. The minimum measurable value of the power meter we used is 1 pW, and the measurements are reliable. The incident energy is stable in the experiment, maintaining the measurement for a long time, the rainbow remains constant, and the outgoing energy is zero, indicating that the light is trapped in the waveguide region. As a comparative experiment, the conclusions were repeatedly calibrated and the energy outflow was determined to be constant to zero over a considerable period of time.

In Fig. 5a, it is obvious that there is a big difference in the normalized optical power, for the TiO₂@PMMA waveguide and the green light metamaterial waveguide. When the opening size is over 32 μm , the power density of the TiO₂@PMMA waveguide is about 0.8 and that of the green light waveguide is about 0.4, where the energy reduction is exactly caused by propagation loss, that is, the loss of the green light waveguide is much larger than that of TiO₂@PMMA waveguide. However, as the opening continues to decrease, light of different incident wavelengths exhibits completely different morphologies. For the measured wavelengths from 473 nm to 671 nm, the energy density decreases; until the opening size of 20 μm , where the energy density of the 589 nm wavelength decreases to 0.2. However, that of 532 nm wavelength behaves differently, and as the opening decreases, the energy density decreases drastically, and gradually decreases to zero as the opening becomes smaller than 24 μm . Actually, this is a manifestation of the fact that the resonant wavelength of the green light metamaterials is close to the input wavelength, and back reflection occurs, which increases the energy density reduction, and finally, there is a complete back reflection, resulting in zero energy outflow. The results of the red light metamaterial waveguide (Fig. 5b) are similar to those of the green light metamaterial waveguide, and for the broadband sample waveguide, Fig.5c illustrates the phenomenon remarkably well. As the broadband resonance, the outgoing energy of various wavelengths is significantly reduced compared with the single-frequency response. For example, for 30 μm opening size and under the same incident energy intensity, the outgoing power of 589 nm wavelength in the green and red resonance is 0.3, while in the case of broadband resonance is reduced to 0.15, for this wavelength is within the range of broadband resonance. Moreover, the measured light at 532 nm, 589 nm, and 632 nm, all experienced zero outgoing power state. It is clear that the slowing down effect of light in the broadband waveguide is enhanced compared to the single frequency waveguide, and the light is imprisoned in a wider frequency range.

Fig. 5d provides a qualitative illustration of this phenomenon. I. In the TiO₂@PMMA waveguide, the incident light undergoes normal GH shift, and produce normal reflection, where part of the energy is consumed in the light-matter interaction and part of the energy exits from the other end of the waveguide. In metamaterial waveguides: II. For non-resonant conditions, similar to in TiO₂@PMMA waveguides, the incident light undergoes only normal GH shifts, the energy of the outgoing light is lower than that of TiO₂@PMMA waveguides due to the higher intrinsic loss of the

metamaterial. III. When resonance occurs, the incident light undergoes negative GH shift and produces normal and back reflections, and the energy of the outgoing light further decreases. IV. When the opening size of the waveguide continues to decrease, i.e., when the angle of incidence decreases further, back reflections are enhanced, normal reflections are weakened, and the energy of the outgoing light becomes even lower. V. When the angle of incidence decreases to the critical value, perfect back reflections of the incident light occur, leading to zero-outgoing-power. These experimental results show that the trap effect associated with group velocity reduction is clearly different from the system's own losses, and that the reduction in energy density is caused by back reflections formed by the resonance state, whereas the zero energy outflow is the result of the complete back reflections. Perfect absorption of metamaterials⁵⁸ has been widely studied⁵⁹, which achieves basically no reflection for electromagnetic waves incident on the surface of metamaterials through the design principle of impedance matching, and at the same time, the refractive index imaginary part of metamaterials is designed as large as possible, so that all the energy of the electromagnetic waves can be absorbed up to 98%. Different from the perfect absorption, this work is to form the back reflection through the anomalous GH effect of the negative refractive material, and to reach the perfect back reflection at the critical state, so that the outgoing energy of the waveguide becomes zero, which exhibits the optical black hole phenomenon.

Fig. 5 | Output power measurement. **a, b** Normalized curves of output light power measured at the outgoing port of the green-light metamaterial waveguide and red-light metamaterial waveguide. The output light power varies with the incoming port d_1 of the waveguide, and in these two waveguides output light power of green and red light were observed to decrease to 0, respectively. **c** Normalized output light power of the broadband sample waveguide. When the light power incident into the waveguide is limited to 31 pW, the output light power of red, yellow, and green light decreases to 0 as the incident port d_1 varies from 20 to 26 μm . **d** Schematic diagram of perfect back reflection. In: Incidence, R: reflection, NGH: Negative GH effect, BR: Back reflection, PBR: Perfect back reflection.

We have rewritten the “**Perfect back reflection: an optical black hole**” section in the revised manuscript: “The 3D wedge-shaped optical waveguides consisting of two planar samples mentioned above are also used for light power measurement, respectively, and the heights of the incoming and outgoing ports are d_1 and d_2 ($d_1 > d_2$,

and d_2 is set as $2.4 \mu\text{m}$ here). We used the setup shown in Fig. 3a and measured the variation of the output optical power from the rear port of the waveguide with the size of the waveguide incoming port d_1 , at a known and constant incident power. The experiments were conducted in an optical darkroom, the effective light power incident into the waveguide was limited to 31 pW and a NOVAVII handheld optical power tester with measurement accuracy of 1 pW was used. Tables 1, 2 and 3 show the test data for the output energy flow of the green, red and broadband materials as well as the $\text{TiO}_2\text{@PMMA}$ waveguide, respectively, which were then normalized by taking the incident light power as the maximum value (31 pW) to obtain Fig. 5a-c. We measured the variation of the outgoing optical power with the size of incoming port d_1 of the waveguide, and the results show the state of the light in the waveguide. It can be found that for a fixed outlet $d_2 = 2.4 \mu\text{m}$, the decreasing incident opening of the waveguide leads to variations in the outgoing optical power, and the light is completely stopped when reaching the critical thickness, and the emitted optical power decreases to 0. When the opening size is increased over a certain thickness, the trapped light escapes rapidly, and the outgoing light power increases and tends to stabilize, and the waveguide is almost without the ability of the loss of light, at which time the loss of light power is due to the inherent loss of the waveguide material (if there is no loss of light in the waveguide, the normalized output optical power should be 1).

The back reflection effect in the light-stopping state was observed experimentally. Fig. 4a–c can illustrate that there is a rainbow effect in the thin film waveguide composed of $\text{Ag/AgCl/TiO}_2\text{@PMMA}$ particles (yellow part), as a result of backward reflection due to the anomalous GH effect, while there is no anomalous GH effect and back reflection occurring in the waveguide composed of $\text{TiO}_2\text{@PMMA}$ particles (grey part), and therefore it cannot cause the rainbow. It is already known that when attempting to slow or stop optical signals in nanoscale structures, the key issue to be addressed is how dissipation losses^{54,55} and backscattering channels^{56,57} affect the light-deceleration ability of these devices. In the waveguide composed of $\text{TiO}_2\text{@PMMA}$ particles, there are dissipation losses and backward scattering, but it is difficult to form back reflection, so the rainbow effect cannot be achieved. “Rainbow” formation in the waveguide composed of $\text{Ag/AgCl/TiO}_2\text{@PMMA}$ particles is precisely due to the back reflection caused by the anomalous GH effect. Fig. 5 reacts to the back reflection phenomenon in metamaterial waveguides and the complete back reflection that occurs when reaching the critical opening state. When the incoming port reaches a critical value, the outgoing energy flow is experimentally observed to be zero, indicating a light-stopping state, and the back reflection can be maintained for a long time under continuous energy input. We call the behavior of this state perfect back reflection: an optical black hole, a state when the outgoing light is fully captured. The minimum measurable value of the power meter we used is 1 pW , and the measurements are reliable. The incident energy is stable in the experiment, maintaining the measurement for a long time, the rainbow remains constant, and the outgoing energy is zero, indicating that the light is trapped in the waveguide region. As a comparative experiment, the conclusions were repeatedly calibrated and the energy outflow was determined to be constant to zero over a considerable period of time.

In Fig. 5a, it is obvious that there is a big difference in the normalized optical power, for the TiO₂@PMMA waveguide and the green light metamaterial waveguide. When the opening size is over 32 μm, the power density of the TiO₂@PMMA waveguide is about 0.8 and that of the green light waveguide is about 0.4, where the energy reduction is exactly caused by propagation loss, that is, the loss of the green light waveguide is much larger than that of TiO₂@PMMA waveguide. However, as the opening continues to decrease, light of different incident wavelengths exhibits completely different morphologies. For the measured wavelengths from 473 nm to 671 nm, the energy density decreases; until the opening size of 20 μm, where the energy density of the 589 nm wavelength decreases to 0.2. However, that of 532 nm wavelength behaves differently, and as the opening decreases, the energy density decreases drastically, and gradually decreases to zero as the opening becomes smaller than 24 μm. Actually, this is a reflection of the fact that the resonant wavelength of the green light metamaterials is close to the input wavelength, and back reflection occurs, which increases the energy density reduction, and finally, there is a complete back reflection, resulting in zero energy outflow. The results of the red light metamaterial waveguide (Fig. 5b) are similar to those of the green light metamaterial waveguide, and for the broadband sample waveguide, Fig.5c illustrates the phenomenon remarkably well. As the broadband resonance, the outgoing energy of various wavelengths is significantly reduced compared with the single-frequency response. For example, for 30 μm opening size and under the same incident energy intensity, the outgoing power of 589 nm wavelength in the green and red resonance is 0.3, while in the case of broadband resonance is reduced to 0.15, for this wavelength is within the range of broadband resonance. Moreover, the measured light at 532 nm, 589 nm, and 632.8 nm, all experienced zero outgoing power state. It is clear that the slowing down effect of light in the broadband waveguide is enhanced compared to the single frequency waveguide, and the light is imprisoned in a wider frequency range. Fig. 5d provides a qualitative illustration of this phenomenon. I. In the TiO₂@PMMA waveguide, the incident light undergoes normal GH shift, and produce normal reflection, where part of the energy is consumed in the light-matter interaction and part of the energy exits from the other end of the waveguide. In metamaterial waveguides: II. For non-resonant conditions, similar to that in TiO₂@PMMA waveguides, the incident light undergoes only normal GH shifts, the energy of the outgoing light is lower than that of TiO₂@PMMA waveguides due to the higher intrinsic loss of the metamaterial. III. When resonance occurs, the incident light undergoes negative GH shift and produces normal and back reflections, and the energy of the outgoing light further decreases. IV. When the opening size of the waveguide continues to decrease, i.e., when the angle of incidence decreases further, back reflections are enhanced, normal reflections are weakened, and the energy of the outgoing light becomes even lower. V. When the angle of incidence decreases to the critical value, perfect back reflections of the incident light occur, leading to zero-outgoing-power. The physical mechanism of the “rainbow” is backward reflection caused by particles in the waveguide. The displacement caused by the negative GH effect increases the actual distance travelled by the wave in the waveguide, which leads to a decrease in the group velocity of the wave packet,

exhibiting the slow wave effect. The state of zero energy outflow, when it occurs, implies that the light is completely captured and stopped at a fixed position. The negative GH shifts measured in Fig. 2 is 300–700 nm, and it can be estimated that the time required for once displacement is about 1–2.33 fs, i.e., the speed of light travelling through the waveguide is slowed down; when the energy flow is zero, that is, light is stopped.

Experiments have shown that the phenomenon of zero energy outflow can occur in negative refraction waveguides under certain conditions. For the waveguide prepared from TiO₂@PMMA, it causes a reduction in the outgoing energy, but the phenomenon of zero-energy-outflow will not occur (Fig. 5). Even for the negative refractive medium waveguide, when the wavelength of the incident light is different from the resonant frequency, a large energy loss can be formed, but no zero-energy-outflow occurs as well (Fig. 5). Obviously, when resonant occurs in the negative refraction frequency region, the interaction between the light and the medium leads to a great dissipation of energy, due to the back reflection of the incident light. And when the critical conditions are reached, the incident energy is completely dissipated, resulting in a phenomenon where the energy is continuously input in the waveguide for a long period of time, but no energy is outflowed. Therefore, we call this state of zero energy outflow as the formation of an energy black hole, i.e., an "optical black hole". Certainly, this phenomenon is fundamentally different from the optical black hole spoken of in astronomy.

These analyses show that the trap effect associated with group velocity reduction is clearly different from the system's own losses, and that the reduction in energy density is caused by back reflections formed by the resonance state, whereas the zero-energy-outflow is the result of the complete back reflections. Perfect absorption of metamaterials⁵⁸ has been widely studied⁵⁹, which achieves basically no reflection for electromagnetic waves incident on the surface of metamaterials through the design principle of impedance matching, and at the same time, the refractive index imaginary part of metamaterials is designed as large as possible, so that all the energy of the electromagnetic waves can be absorbed up to 98%. Different from the perfect absorption, this work is to form the back reflection through the anomalous GH effect of the negative refractive material, and to reach the perfect back reflection at the critical state, so that the outgoing energy of the waveguide becomes zero, which exhibits the optical black hole phenomenon. The present scheme simultaneously allows for high in-coupling efficiencies and broadband, room-temperature operation. The underlying physics of slowing light with negatively refracting media in which light experiences a negative electromagnetic environment in the core layer of the waveguide, forming a negative GH effect^{1, 12}, subverts the conventional slow-wave approach based on resonance or periodic configurations above the diffraction limit, and highlights the interaction of waves with matter.”

58. Landy, N. I., Sajuyigbe, S., Mock, J. J., Smith, D. R., Padilla, W. J. Perfect metamaterial absorber. *Phys. Rev. Lett.* **100**, 207402 (2008).

59. Liu, X., Xia, F., Wang, M., Liang, J., Yun, M. Working Mechanism and Progress of Electromagnetic Metamaterial Perfect Absorber. *Photonics* **10**, 205 (2023).

4. Finally, it seems to me that the information provided does not allow the reproduction of the results presented. It would be useful to propose a more complete description of the methods used to synthesize these meta-structures, as well as a more complete description of the experimental set-ups carried out.

Response 4: Thank you for your suggestion. We have added information on the structure, morphology, parameters, and optical properties of the metamaterials in Fig. 1, with a critical description in the first paragraph of the **Result** section in the revised manuscript: “Fig. 1a shows the schematic diagram of the construction of axially varying heterogeneous film waveguides for slowing down light, based on proposed broadband omnidirectional visible metamaterials. Using the monochrome particles red and green⁴² and the combination of eight kinds of particles⁴⁶, monochrome film and broadband film can be prepared respectively (see Methods). The broadband planar film samples are assembled automatically by a list of narrowband, ultra-low loss isotropic meta-cluster system without selection, covering the main bands from green light to red light. The broadband waveguide consists of two planar samples (see Methods).

As has been reported, we have invented an ultralow loss isotropic metamaterial in the visible spectrum⁴², and the ball-thorn-shaped metamaterial cluster model consist of a spherical kernel and many protruding rods (Fig.1b–c shows green light and red light models). In the simulation, both the kernel and rods are made of TiO₂ coated by Ag of 1 nm in thickness. 600 identical rods with cross-sectional diameter of 15 nm are uniformly distributed around the surface of a kernel. l represents the diameter of the meta-cluster, r is the radius of the spherical kernel, and P refers to the lattice constant of the meta-cluster, the meta-cluster is fully immersed in polymethyl methacrylate (PMMA). The structural parameters of red-light model are $l = 640$ nm, $r = 215$ nm, and $P = 670$ nm, and that of green-light model are $l = 530$ nm, $r = 165$ nm, and $P = 560$ nm. Adjusting l , r , and P according to scaling rule enables the model to be applicable to different working frequency bands. The cluster is named Ag/AgCl/TiO₂@PMMA, and the ball-thorn-shaped meta-clusters with symmetrical structure consisting of the dielectric and its surface dispersed super-thin silver layer have replaced the lithographically defined meta-atoms in existing negative-index metamaterials (NIMs); it is found that the discrete super-thin silver layer produced by the photoreduction method can generate plasmon resonance when excited by electromagnetic waves, thereby achieving the performance of metamaterials. The significant reduction in silver coating thickness provides the physical basis for the decreased joule heating and the realization of ultralow losses. Subsequently, we successfully obtained a novel broadband omnidirectional metamaterial (Fig. 1d shows the unit of broadband omnidirectional meta-clusters system) randomly assembled by a list of narrowband, omnidirectional, and ultralow loss meta-cluster system using a bottom-up approach⁴⁶. Fig. 1e shows the TEM images of the particles that resonate in the green (left) and red (right) light spectrum, revealing a classic kernel–shell structure. The TEM images show

that the size of the ball-thorn-shaped Ag/AgCl/TiO₂ particle is approximately 500–700 nm, and the thickness of the PMMA shell is nearly 20–30 nm. The negative refraction for red sample occurs at around 610–640 nm, and the minimum refractive index is –0.41 at 630 nm; the negative refraction for green sample occurs at around 520–550 nm, and the minimum refractive index is about –0.30 at 532 nm. Fig. 1f shows the SEM image of broadband film sample experimentally obtained by disordered self-assembly of eight kinds of single-frequency meta-clusters. For the first time, the negative refractive index of 490 nm–730 nm band (Fig. 1g) and broadband inverse Doppler effect across most of the visible spectrum was observed in the proposed broadband metamaterials. Simulated refractive index of the green and red metamaterials and approximations of the real part of the refractive index calculated with the Kramers–Kronig integral⁵¹ are shown in the Fig. 1h, and they basically satisfy the KK relationship. In addition, related study¹ has shown that that the negativity in the real part of the effective refractive index enables the deceleration of light in the layers of the heterostructure. On these bases, the research in this paper can be conveniently carried out.”

51. Szabo, Z. Closed Form Kramers-Kronig Relations to Extract the Refractive Index of Metamaterials. *IEEE Trans. Microwave Theory Tech.* **65**, 1150-1159 (2017).

Fig. 1 | Conceptual illustration of the axially varying heterogeneous waveguides based on proposed broadband omnidirectional visible metamaterials. **a** Schematic diagram of axially varying heterogeneous waveguides composed of broadband planar film samples. **b** Green light meta-cluster model with parameters of $l = 530$ nm, $r = 165$ nm, and $P = 560$ nm (left Figure), the cluster is composed of a spherical core and many prominent rods. When the response band light ray incident, the meta-clusters will occur negative refraction effect. The profile current distribution perpendicular to the external magnetic field and the 1/8 model profile (right picture). **c** Red-light meta-cluster model with parameters of $l = 640$ nm, $r = 215$ nm, and $P = 670$ nm. **d** Cluster unit of broadband omnidirectional meta-clusters system; it is composed of 8 clusters with response bands

of 490 nm, 500 nm, 540 nm, 570 nm, 600 nm, 640 nm, 680 nm, and 700 nm, respectively. **e** TEM images of green-light (left) and red-light (right) particles. The size of the ball-thorn-shaped Ag/AgCl/TiO₂ particle is approximately 500–700 nm, and the thickness of the PMMA shell is nearly 20–30 nm. **f** SEM image of the monolayer film of the broadband metamaterial, which is obtained by disordered self-assembly of eight kinds of single-frequency particles. **g** Refractive index curve measured for broadband sample S_B, green-light sample S_G, and red-light sample S_R. S_{Bsim} is the simulated value of the broadband sample. **b**, **c** and **e** cite from Ref. 42; **d**, **f**, and **g** cite from Ref. 46. **h** Simulated refractive index of the green and red metamaterials and approximations of the real part of the refractive index calculated with the Kramers–Kronig integral.

Moreover, we have added the methods for synthesizing meta-clusters and experimental setups carried out in the **Methods** section as follows: “**Preparation and characterisation of meta-clusters**

The Ag/AgCl/TiO₂@PMMA meta-cluster particles corresponding to red-light and green-light are prepared using the solvothermal synthesis method⁴². In order to solve the problem of the coating of nano-silver layer of ball-thorn-shaped clusters, AgCl is firstly formed by mixing a certain amount of AgNO₃ into TiCl₄ during the process of preparing the TiO₂ rods. After a photoreduction method, AgCl further disintegrates into elemental chlorine and metallic silver. The latter precipitates on the outer surface of the ball-thorn-shaped structure to form the discrete silver distribution about 1 nm. Next, these agglomerated particles are immersed in PMMA and illuminated to form the Ag/AgCl/TiO₂@PMMA particles.

Eight kinds of Ag/AgCl/TiO₂@PMMA (meta-cluster) nanoparticles were prepared by solvothermal synthesis from bottom to top⁴⁶. The transmission electron microscopy (TEM) images of the meta-cluster particles that resonate in the red-, yellow-, and green-light spectra reveal a classic kernel (AgCl/TiO₂)–shell (PMMA) structure. The images confirm the presence of PMMA filling between different nanorods, and the thickness of the PMMA shell is nearly 20–30 nm. According to the idea of multi-frequency composite, the meta-cluster particle system in the wide band of visible light was prepared.

The meta-cluster particle set was characterized by the optical micrograph and scanning electron microscopy (SEM) images, the transmission electron microscopy (TEM) images, the local high-angle annular dark-field imaging scanning TEM (HAADF-STEM) images, X-ray diffraction (XRD) patterns, Ultraviolet–visible–near infrared (UV-VIS-NIR) absorption spectra^{42,46}.”

REVIEWER COMMENTS

Reviewer #1 (Remarks to the Author):

I have read the revised version of the paper, and the authors replies to the referees.

From my perspective, the authors have done a good job in addressing all of the key points that I previously asked them. In particular, it is now clear that the authors are the first to observe the crucial, negative Goos-Hanchen shift (in the visible regime), as well as the associated negative refractive index for the 'rainbow' effect, whose real and imaginary parts obey Kramers-Kronig relations (i.e., describe a causal, physical quantity, the refractive index n , with $\text{Re}\{n\} < 0$). The authors are now also clear that they do observe back-reflections, as with previous experiments on this effect.

All in all, this is a very delicate experiment, observing for the first time so clearly the 'rainbow' effect at visible frequencies, with all of its main features, and with several accompanying side-results enhancing one's confidence that the effect has indeed been observed as proclaimed. The observed effect is also, clearly, broadband - which is very rare for 'slow light' waves.

For the above reasons, I am now confident that this work makes an interesting and important new contribution to the wider fields of metamaterials and 'rainbow trapping' / slow light, as to merit publication in NCOMMs as is.

Reviewer #2 (Remarks to the Author):

In the revised version, I understood the used metamaterial structure and its optical characteristics. But still I cannot see why the authors claim its low loss although it exhibits a large extinction coefficient from the metal. I cannot find any evidence of low loss or I wonder what the definition of low loss is.

Moreover, since this study is based on many previous works, as I indicated previously, the results should reduce its novelty, which is not suitable for publishing in Nature

Communications. Otherwise, its explanations and discussion are very long, complicated, and redundant.

In the revised version, the slow light and stopping of light effects are not still clear. It seems to me just waveguiding in a tapered free space sandwiched by two DOE-like plates. The colorful output called rainbow effect looks just a result after multiple diffraction. This amber color was caused by the disordering of pure rainbow, wasn't it?

Although it comes from a negative GE shift, the result is not so different and not attractive as it is. I do not believe that the device achieved effective accumulation optical energy in the device. The stopping of light must be just the reflection at a critical point after the multi-diffraction.

In any case, this study lacks simple, careful, and solid comparison between experiment and theory.

Reviewer #3 (Remarks to the Author):

A significant amount of work has been provided by the authors to answer my questions and complete the manuscript.

I find that in its new form, the paper provides a much clearer presentation of its contributions in relation to the literature, as well as experimental elements and illustrations that can more easily guide the reader to understand the phenomena involved and form a solid analysis of the subject.

In line with a previous remark that I had identified as problematic, the technicality involved in the development of this experimental proof seems to me to limit the possible reproduction of the work from. In this new version, the authors provide more detailed information on the set-up of their experiments, as well as data from measurements.

My one and only reservation in recommending the publication of this work lies in the fact

that I am still cautious about reproducing it from the data available in the Methods section alone. Putting myself in the role of an experimentalist, a lot of information is missing to reproduce the meta-clusters described in this work and in previous publications. I suspect that, beyond the sole concern of compactness, mastering the production of these particles represents an academic and/or commercial challenge on which the authors wish to maintain a certain lead by keeping production secrets.

In view of the quality of the work presented, I shall leave the recommendation of this publication to the discretion of the editors, who will be in a better position to judge whether reproducibility is a barrier to publication in Nature Communications.

After careful examination of the reviewers' comments, we have responded to the comments point by point (in red) in this letter and made the corresponding modifications (in blue) in the revised manuscript. The followings are the responses in detail:

Reviewer #1 (Remarks to the Author):

I have read the revised version of the paper, and the authors replies to the referees.

From my perspective, the authors have done a good job in addressing all of the key points that I previously asked them. In particular, it is now clear that the authors are the first to observe the crucial, negative Goos-Hanchen shift (in the visible regime), as well as the associated negative refractive index for the 'rainbow' effect, whose real and imaginary parts obey Kramers-Kronig relations (i.e., describe a causal, physical quantity, the refractive index n , with $\text{Re}\{n\} < 0$). The authors are now also clear that they do observe back-reflections, as with previous experiments on this effect.

All in all, this is a very delicate experiment, observing for the first time so clearly the 'rainbow' effect at visible frequencies, with all of its main features, and with several accompanying side-results enhancing one's confidence that the effect has indeed been observed as proclaimed. The observed effect is also, clearly, broadband - which is very rare for 'slow light' waves.

For the above reasons, I am now confident that this work makes an interesting and important new contribution to the wider fields of metamaterials and 'rainbow trapping' / slow light, as to merit publication in NCOMMs as is.

Response: Thank you for your positive comments about our research.

Reviewer #2 (Remarks to the Author):

In the revised version, I understood the used metamaterial structure and its optical characteristics. But still I cannot see why the authors claim its low loss although it exhibits a large extinction coefficient from the metal. I cannot find any evidence of low loss or I wonder what the definition of low loss is.

Moreover, since this study is based on many previous works, as I indicated previously, the results should reduce its novelty, which is not suitable for publishing in Nature

Communications. Otherwise, its explanations and discussion are very long, complicated, and redundant.

In the revised version, the slow light and stopping of light effects are not still clear. It seems to me just waveguiding in a tapered free space sandwiched by two DOE-like plates. The colorful output called rainbow effect looks just a result after multiple diffraction. This amber color was caused by the disordering of pure rainbow, wasn't it?

Although it comes from a negative GE shift, the result is not so different and not attractive as it is. I do not believe that the device achieved effective accumulation optical energy in the device. The stopping of light must be just the reflection at a critical point after the multi-diffraction.

In any case, this study lacks simple, careful, and solid comparison between experimnt and theory.

1. In the revised version, I understood the used metamaterial structure and its optical characteristics. But still I cannot see why the authors claim its low loss although it exhibits a large extinction coefficient from the metal. I cannot find any evidence of low loss or I wonder what the definition of low loss is.

Response 1: Thank you for your review. I'm sorry we didn't explain it clearly last time. As a continuous series of work on optical metamaterials, we introduced the physical model and experimental results for reducing metamaterial loss in detail in Ref. 42, containing detailed evidence for low loss: (Zhao, J., et al. *Nanophotonics* 11, 2953-2966 (2022))]:

1. Physical model (Excerpt from Ref.42)

Inspired by the ciliated cell structure, we created a ball-thorn-shaped metamaterial cluster (meta-cluster) model consists of a spherical kernel and many protruding rods (Figure 1b, left picture) as analogous to the cilium-cell structure found in nature. Both the kernel and rods are made of TiO₂ coated by Ag of 1 nm in thickness. In fact, each ball-thorn-shaped meta-cluster can be regarded as composed of 1200 meta-atoms: U-shaped split-ring and equivalent wires distributed evenly and symmetrically along the space. These meta-atoms geometrical size is much smaller than the wavelength, and it is their independent resonance that forms the meta-cluster resonant response outfield (Figure 1b, right picture). Theoretical calculations have

shown that the plasma resonance can be formed with a silver layer height of only two or three atomic layers. When the response band light ray incident, the meta-clusters will occur negative refraction effect. Our cluster design is independent of the previously widely used meta-atom cell design, this model greatly reduces the silver coating thickness required for achieving high FOM. It is indeed this meta-clusters NIMs significant reduction in silver coating thickness that provides the physical basis for the decreased joule heating and thus the realization of ultra-low losses, and the resulting FOM is nearly an order of magnitude greater than the state-of-arts.

[FIGURE REDACTED]

Figure1: Behavior of the meta-clusters structure. (a) Schematic of the biological cilium-cell. (b) Green light wave band meta-cluster model (left Figure), the cluster is composed of a spherical core and many prominent rods. When the response band light ray incident, the meta-clusters will occur negative refraction effect. The profile current distribution perpendicular to the external magnetic field and the 1/8 model profile (right picture). The protruding rods and epidermic connections form "U" shaped effective units, the negative permeability is generated when the annular current and the external magnetic field interact with each other, which is regarded as the basic effective units of the meta-cluster (there are ~1200 units). The whole meta-cluster is in the PMMA environment background. The model profile shows that the model is filled with TiO₂ medium (white area) and the epidermis is covered with a very thin layer of Ag (green part, 1nm). (c) Transmission (solid line) and reflection (dot-dash line) coefficient for the red-light meta-clusters with $l = 640$ nm, $r = 215$ nm, and $P = 670$ nm. (d) The effective parameters for the red-light meta-clusters retrieved from the coefficients in (c). (e) Effective refractive indices for the red-light meta-clusters (red line) and the green-light meta-clusters (green line) with $l = 530$ nm, $r = 165$ nm, and $P = 560$ nm. (f) FOM curves of the meta-clusters resonating at the red-light (red dotted line) and green-light (green dotted line), respectively. (g) FOM of the red-light meta-clusters structure as a function of Ag layer thickness t_{Ag} . (h) FOMs of fishnet structures at different Ag layer thickness. Black square and red circle lines represents the results obtained using the Ag-Al₂O₃-Ag fishnet structure whose geometrical parameters refer to the published work [17] and blue triangle and green inverted-triangle lines represents the results obtained using the Ag-MgF₂-Ag fishnet structure whose geometrical parameters refer to the published work [19].

2. Experimental result (Excerpt from Ref.42)

The Ag/AgCl/TiO₂@PMMA meta-cluster particles corresponding to red-light and green-light are prepared using the solvothermal synthesis method. In order to solve the problem of the coating of nano-silver layer of ball-thorn-shaped clusters, AgCl is firstly formed by mixing a certain amount of AgNO₃ into TiCl₄ during the process of preparing the TiO₂ rods. After a photoreduction method, AgCl further disintegrates into elemental chlorine and metallic silver. The latter precipitates on the outer surface of the ball-thorn-shaped structure to form the discrete silver distribution about 1 nm. Next, these agglomerated particles are immersed in PMMA and illuminated to form the Ag/AgCl/TiO₂@PMMA particles. The Ag layer can

generate plasmon resonance when excited by electromagnetic waves, thereby achieving the performance of metamaterials.

Figure 3: Characterization of the ball-thorn-shaped composite particle. (a), (b) Local HAADF-STEM images of the Ag/AgCl/TiO₂ particles, silver appears as surface dispersion distribution on the ball-thorn surface. (c) TEM image of Ag/AgCl/TiO₂ particle; (d), (e) the corresponding elemental mapping of Ti and Ag, respectively. (f) XPS analysis of Ag 3d in Ag/AgCl/TiO₂@PMMA.

3. Experimental evidence for low loss:

Consistent with studies in this field, we use the figure of merit ($FOM = -\text{Re}(n)/\text{Im}(n)$ for $\text{Re}(n) < 0$, and $\text{Re}(n)$ and $\text{Im}(n)$ are the real and imaginary parts of the refractive index, respectively) to evaluate the loss of metamaterials, with higher FOM values predicting lower loss. In Ref. 42, we comprehensively compared the FOM values of meta-cluster with those of other works, and our obviously superior FOM values supported the low loss characteristics of the proposed meta-cluster [see the following **Table** excerpted from Ref. 42. Furthermore, Ref. 46 describes the FOM of the broadband sample.

[TABLE REDACTED]

Table 1 Method and performance comparison of visible metamaterials (excerpted from Ref. 42)

The references involved are ordered in Ref. 42:

- [16] J. Valentine, S. Zhang, T. Zentgraf, et al., "Three-dimensional optical metamaterial with a negative refractive index," *Nature*, vol. 455, p. 376-9, 2008.
- [17] S. Xiao, U. K. Chettiar, A. V. Kildishev, V. P. Drachev, V. M. Shalaev. "Yellow-light negative-index metamaterials," *Opt. Lett.*, vol. 34, p. 3478-3480, 2009.
- [18] G. Dolling, M. Wegener, C. M. Soukoulis, S. Linden. "Negative-index metamaterial at 780 nm wavelength," *Opt. Lett.*, vol. 32, p. 53-55, 2007.
- [19] Y. Liang, Z. Yu, N. Ruan, Q. Sun, T. Xu. "Freestanding optical negative-index metamaterials of green light," *Opt. Lett.*, vol. 42, p. 3239-3242, 2017.
- [39] M. Gómez-Castaño, J. L. Garcia-Pomar, L. A. Pérez, S. Shanmugathan, S. Ravaine, A. Mihi. "Electrodeposited Negative Index Metamaterials with Visible and Near Infrared Response," *Adv. Opt. Mater.*, vol. 8, p. 2000865, 2020.
- [40] J. Yao, Z. Liu, Y. Liu, et al., "Optical negative refraction in bulk metamaterials of nanowires," *Science*, vol. 321, p. 930-930, 2008.
- [51] S. Xiao, V. P. Drachev, A. V. Kildishev, et al., "Loss-free and active optical negative-index metamaterials," *Nature*, vol. 466, p. 735-8, 2010.
- [52] U. K. Chettiar, A. V. Kildishev, H. K. Yuan, W. Cai, V. M. Shalaev. "Dual-Band Negative Index Metamaterial: Double-Negative at 813 nm and Single-Negative at 772 nm," *Opt. Lett.*, vol. 32, p. 1671-1673, 2007.
- [53] C. Garcia-Meca, J. Hurtado, J. Marti, A. Martinez, W. Dickson, A. V. Zayats. "Low-loss multilayered metamaterial exhibiting a negative index of refraction at visible wavelengths," *Phys. Rev. Lett.*, vol. 106, p. 067402, 2011.
- [54] Y. J. Jen, C. H. Chen, C. W. Yu. "Deposited metamaterial thin film with negative refractive index and permeability in the visible regime," *Opt. Lett.*, vol. 36, p. 1014-1016, 2011.

As can be seen from the above, theoretical calculations have shown that this ball-thorn-shaped meta-cluster reduces the thickness of the metallic silver layer required to generate local plasmon resonance (LPR) from the micrometer scale to the height of two or three atomic layers. The experiment further shown that plasmonic resonance can be achieved by replacing a continuously distributed silver layer with a discretely distributed metallic silver layer. We achieved the first experimental determination of negative refraction in a three-dimensional visible metamaterial sample and confirmed the low loss of this material⁴².

The following sentences have been added to the first paragraph of the "Results" section in the revised manuscript: "In order to achieve low losses, a ball-thorn-shaped cluster model is

applied, and theoretical calculations show that this meta-cluster reduces the thickness of the metallic silver layer required to generate local plasmon resonance (LPR) from the micrometer scale to the height of two or three atomic layers. The experiment further shown that plasmonic resonance can be achieved by replacing a continuously distributed silver layer with a discretely distributed metallic silver layer. We have achieved the first experimental determination of negative refraction in a three-dimensional visible metamaterial sample and confirmed the low loss of this material⁴².”

In addition, we add this description to the second paragraph of the “**Results**” section in the revised manuscript: “The lowest figure of merit ($FOM = -\text{Re}(n)/\text{Im}(n)$) was 6.7 at 538 nm and the highest value was 13 at 592 nm, which predicts low losses⁴⁶.”

2. Moreover, since this study is based on many previous works, as I indicated previously, the results should reduce its novelty, which is not suitable for publishing in Nature Communications. Otherwise, its explanations and discussion are very long, complicated, and redundant.

Response 2: Thank you for your review. Optical metamaterial, especially visible band metamaterial, is one of the most important and difficult target in the field of metamaterials. The orientation and resonance of the artificial structure unit give rise to three endogenous restrictive problems of directional response, high loss and narrowband in the metamaterial, which turn out to be intractable challenges. We have made breakthroughs on these questions in previous research:

I. Proposing symmetrically structured spherical ball-thorn-shaped meta-cluster, consisting of the dielectric and its surface dispersed super-thin silver layer (Ref. 42). Inspired by the ciliated cell structure, we created a ball-thorn-shaped metamaterial cluster (meta-cluster) model with a symmetric structure, consisting of the dielectric and its surface dispersed super-thin silver layer, to replace the lithographically defined meta-atoms. Theoretical calculations have shown that the plasma resonance can be formed with a silver layer height of only two or three atomic layers. When the response band light ray incident, the meta-clusters will occur negative refraction effect. The experiment proves that the discrete super-thin silver layer can generate negative refraction. Our cluster design is independent of the previously widely used meta-atom

cell design, this model greatly reduces the silver coating thickness required for achieving high figure of merit (FOM). It is indeed this meta-clusters NIMs significant reduction in silver coating thickness that provides the physical basis for the decreased joule heating and thus the realization of ultra-low losses, and the resulting FOM is nearly an order of magnitude greater than the state-of-arts. The proposed ball-thorn-shaped meta-clusters structure breaks through the dilemma of whether to use noble metals in engineering visible light meta-atom NIMs. In addition, spherically symmetric cluster units directly solve the anisotropy problem of meta-atom structure.

II. Proposing broadband omnidirectional metamaterials self-assembled by disordered meta-cluster particles with different scales. (Ref. 46). A method for assembling broadband omnidirectional ultra-low loss visible spectral metamaterials is proposed, based on the conclusions of weak interactions previously found in microwave and acoustic. Theoretical and experimental evidence demonstrates that the broadband metamaterials self-assembled by disordered meta-cluster particles at different scales can operate efficiently over a wide frequency range of omnidirectionally incident visible light and exhibit an automatic frequency selection response. The automatic frequency selection property exhibited by weak interaction provides the physical basis for the bottom-up assembly of meta-clusters.

Our previous studies breaking through the difficulties of high losses, orientation, and narrowband in the field of metamaterials, opening the door to the preparation of three-dimensional broadband omnidirectional visible spectral ultralow-loss metamaterials, and paving the way for the wide application of optical frequency metamaterials.

Further, the present work is the first experimental article to address the problem of slowing down and stopping the speed of light in metamaterials in the visible band, its novelty is mainly reflected in the abstract and conclusion, listed as follows:

1. Using the prepared metamaterials, we observed the negative Goos-Hänchen effect in broadband samples across the entire visible spectrum.
2. We discovered, for the first time, an amber rainbow ribbon and an optical black hole due to perfect back reflection in optical waveguides, where little light leaks out.
3. Not only does the amber rainbow ribbon effect show an automatic frequency selection response, as predicted by single frequency theoretical models and confirmed by experiments, it also shows spatial periodic regulation, resulting from broadband

omnidirectional visible metamaterials prepared by disordered assembly systems.

4. This work overcomes the great difficulty in trapping broadband visible light by successfully solving the long-standing challenge regarding energy leakage.

In addition, we have added a comparative illustration in the last paragraph of the “**Perfect back reflection: an optical black hole**” section to emphasize the novelty of the work: “Recently, the rainbow trapping has been combined with topological photonics to realize topological rainbows³¹⁻³³ in both Hermitian and non-Hermitian^{60,61} cases, and to develop towards the study of higher-order angular states⁶², broadband, and even multiple rainbows⁶³. In addition, broadband “fractal” rainbow trapping⁶⁴ has been obtained by combining gradient metamaterial-based acoustic waveguides with fractal spectroscopy. Our work challengingly chose the visible metamaterial platform, relying on fabricated 3D isotropic metamaterials, to achieve not only single rainbow trapping in the visible band, but also to discover the amber rainbow phenomenon in a wider bandwidth for the first time. In comparison, the claimed multiple rainbows in topological photonics platform are found to be realized only relying on simulation-designed waveguide beam-splitting paths, which are analogous to the case that multiple light paths produce multiple rainbows (many-to-many); our work experimentally observes the amber rainbow in single light path (one-to-many), which is the result of spatially periodic modulation from weak interactions of disordered assembly metamaterials system, providing a new paradigm different from that of waveguide beam-splitting paths.”

3. In the revised version, the slow light and stopping of light effects are not still clear. It seems to me just waveguiding in a tapered free space sandwiched by two DOE-like plates. The colorful output called rainbow effect looks just a result after multiple diffraction. This amber color was caused by the disordering of pure rainbow, wasn't it?

Response 3: Thank you for your review. The principle of trapped rainbow was first proposed in the Ref. 12 ['Trapped rainbow' storage of light in metamaterials. *Nature* 450, 397-401 (2007)]. Subsequently, the theory has been gradually enriched [1. Ultraslow waves on the nanoscale. *Science* 358, eaan5196 (2017)] and validated by numerous studies [15. 'Rainbow' Trapping and Releasing at Telecommunication Wavelengths. *Phys. Rev. Lett.* 102, 056801 (2009), 17.

Plasmonic Rainbow Trapping Structures for Light Localisation and Spectrum Splitting. *Phys. Rev. Lett.* 107, 207401 (2011), 54. Completely Stopped and Dispersionless Light in Plasmonic Waveguides. *Phys. Rev. Lett.* **112**, 167401 (2014).]. In the literature^{1,12}, such a waveguide with the slowly, spatially varying negative-index heterostructure is a necessary condition to produce trapped rainbow. According to the theory, the “two DOE-like plates” provide a gradual effective waveguide thickness, and at the critical thickness, the light will not be able to propagate further downward and will be effectively trapped inside the left-handed heterostructure. Each frequency component of the wave packet is stopped at a different guide thickness, leading to the spatial separation of its spectrum and the formation of a ‘trapped rainbow’. Since the theoretical work is almost complete, our study is focused on obtaining relevant experimental results. In this manuscript, trapped rainbow results for isotropic metamaterial waveguides identical to theoretical predictions are observed for the first time, and in particular, amber rainbows for broadband metamaterials produced by back reflection are obtained and optical black holes formed by perfect back reflection are obtained: the phenomenon of stopping light, which can be supported by our detailed experimental data.

Regarding the amber rainbow, this is a new phenomenon found in our broadband omnidirectional visible metamaterials prepared by disordered assembly systems. Not only does the amber rainbow ribbon effect show an automatic frequency selection response, as predicted by single frequency theoretical model and confirmed by experiments, it also shows spatial periodic regulation. The waveguides corresponding to the amber rainbow are experimentally demonstrated to be capable of slowing down or even stopping light over a wider frequency range (Fig. 5). The amber rainbow originates from weak interactions; Disorderly assembled particles offer ease of preparation, but this property arises from the intrinsic weak interactions of the meta-clusters and is not due to multiple diffraction. For example, there is no such “diffraction phenomenon” in the negative refraction and Doppler effect experiment in disordered assembled broadband samples, but the disordered distribution system still exhibits frequency selectivity. Therefore, the physical nature of amber rainbow originates from the weak interactions of metamaterial units, which is an intrinsic property completely different from the strong interaction system such as photonic and phononic crystals.

The following description has been added to the “**Perfect back reflection: an optical**

black hole” section in revised manuscript: “Our experimental results confirm the frequency-selective rainbow phenomenon due to the negative GH effect of waveguide composed of isotropic metamaterial in the visible band; In particular, we obtain amber rainbows of broadband metamaterials generated by back reflection from weakly interacting systems and observe optical black holes formed by perfect back reflection: the phenomenon of stopping light.”

4. Although it comes from a negative GH shift, the result is not so different and not attractive as it is. I do not believe that the device achieved effective accumulation optical energy in the device. The stopping of light must be just the reflection at a critical point after the multi-diffraction.

Response 4: The speed of light is the fastest known speed of propagation in nature, and slowing it down or even stopping it has attracted considerable attention. The literature cited in the text attempts to fulfill a portion of the research for this goal [Ref.1. Ultraslow waves on the nanoscale. *Science* **358**, eaan5196 (2017); Ref. 12. 'Trapped rainbow' storage of light in metamaterials. *Nature* **450**, 397-401 (2007); Ref. 15. 'Rainbow' Trapping and Releasing at Telecommunication Wavelengths. *Phys. Rev. Lett.* **102**, 056801 (2009); Ref. 17. Plasmonic Rainbow Trapping Structures for Light Localisation and Spectrum Splitting. *Phys. Rev. Lett.* **107**, 207401 (2011); Ref. 54. Completely Stopped and Dispersionless Light in Plasmonic Waveguides. *Phys. Rev. Lett.* **112**, 167401 (2014).], and these can reflect its attractiveness. However, slowing down light with metamaterials in the visible band has always been a challenge. As mentioned before, we have prepared broadband metamaterials in the visible band, and the first determination of the broadband negative GH effect in three-dimensional visible metamaterials provides the basis for realizing this prediction. The key element in the realization of slowing down light here is the slowing down of the group speed of light due to the negative GH effect, which is not necessarily the effective accumulation of light energy in the device. As for stopping light, we give a definite conclusion with experimental measurements, and a careful analysis of the theoretical model is given to illustrate the process of reaching perfect back reflection from back reflection. It is true that the phenomenon of multiple diffraction exists, but the key point is the perfect back reflection that occurs when

critical conditions are reached, the relevant elements of which are carefully described in the main text.

5. In any case, this study lacks simple, careful, and solid comparison between experiment and theory.

Response 5: Thank you for your suggestions. Since much theoretical work has been published in the literature, our study focuses on realizing these predictions and the relevant experimental measurements are targeted in accordance with the theoretical results. For example, the metamaterial wedge shaped waveguide with negative GH effect shows the rainbow effect, in which light of different frequencies stopping at different positions; as the opening size of the waveguide decreases gradually, the rainbow shows a change from the short wavelength of blue to the long wavelength of red; when the opening size of the waveguide decreases to a certain degree, the energy flow appears to be zero; the behaviors are the same regardless of whether the particles of the red or green wavelength, which is in line with the theoretical prediction of Ref. 12. The experiments now further show the amber rainbows and the formation of black hole phenomena with zero energy flow in broadband metamaterial waveguide. This is the result of spatially periodic modulation of multiple rainbows by the disordered assembly metamaterials system and the generation of perfect back reflections, which greatly enhance light-matter interactions and exceeds the predictions of previous theories, revealing that the experimental results encompass a more complex and profound physical mechanism that needs to be further explored and will drive the development of a more profound theoretical system.

The following description was added to the last paragraph of the “**(b) 3D wedge-shaped optical waveguide consisting of two planar samples**” section in the revised manuscript: “Overall, the aforementioned experimental results show that the metamaterial wedge shaped waveguide with negative GH effect shows the rainbow effect, in which light of different frequencies stopping at different positions; moreover, as the opening size of the waveguide decreases gradually, the rainbow shows a change from the short wavelength of blue to the long wavelength of red; when the opening size of the waveguide decreases to a certain degree, the phenomenon of zero energy flow occurs; the behaviors are the same regardless of whether the

particles of the red or green wavelength, which is in line with the theoretical prediction of Ref. 12. The experiments now further show the amber rainbows and the formation of black hole phenomena with zero energy flow in broadband metamaterial waveguide. This is the result of spatially periodic modulation of multiple rainbows by the disordered assembly metamaterials system and the generation of perfect back reflections, which greatly enhance light-matter interactions and exceeds the predictions of previous theories, revealing that the experimental results encompass a more complex and profound physical mechanism that needs to be further explored and will drive the development of a more profound theoretical system.”

Reviewer #3 (Remarks to the Author):

A significant amount of work has been provided by the authors to answer my questions and complete the manuscript.

I find that in its new form, the paper provides a much clearer presentation of its contributions in relation to the literature, as well as experimental elements and illustrations that can more easily guide the reader to understand the phenomena involved and form a solid analysis of the subject.

In line with a previous remark that I had identified as problematic, the technicality involved in the development of this experimental proof seems to me to limit the possible reproduction of the work from. In this new version, the authors provide more detailed information on the set-up of their experiments, as well as data from measurements.

My one and only reservation in recommending the publication of this work lies in the fact that I am still cautious about reproducing it from the data available in the Methods section alone. Putting myself in the role of an experimentalist, a lot of information is missing to reproduce the meta-clusters described in this work and in previous publications. I suspect that, beyond the sole concern of compactness, mastering the production of these particles represents an academic and/or commercial challenge on which the authors wish to maintain a certain lead by keeping production secrets.

In view of the quality of the work presented, I shall leave the recommendation of this publication to the discretion of the editors, who will be in a better position to judge whether reproducibility is a barrier to publication in Nature Communications.

Response: Thank you for your positive comments on our revised version. The only concern of the reviewer was whether the method we provided in the **Methods** section could reproduce the meta-clusters. This is a very important issue, and the reliability comes from three aspects: reproducibility of sample preparation, stability of sample placement over time, and reproducibility of performance testing. Regarding sample reproducibility, in fact, we have reported a comprehensive and detailed preparation method in our previous work (Ref. 42, 46), which contains the complete steps of the preparation. To avoid too much repetition, only a brief description is given in this paper.

The following is excerpted from the Methods section of Ref. 42 (Zhao, J., et al. *Nanophotonics* 11, 2953-2966 (2022).):

“5.1 Preparation of the meta-cluster particles.

First, Ball-thorn-shaped AgCl/TiO₂ particles preparation. The titanium tetrachloride (TiCl₄) is added dropwise to deionized water (analytical reagent) under ice bath to prepare a 38.5 wt% solution. The silver nitrate (AgNO₃, analytical reagent) is dissolved in deionized water to prepare a solution with a concentration of 0.0395 g/mL. The AgNO₃ solution is added to the tetrabutyl titanate (TBT) and toluene mixture and stirred for 30 min. A certain amount of TiCl₄ solution is also added and stirred for 1 h. The mixture is transferred to a Teflon-lined autoclave. The reactor is placed in a constant-temperature drying oven (101A-1E) at 150°C for 24 h. The obtained product is washed several times with absolute ethanol (EtOH, analytical reagent), and then dispersed in ethanol for use or filtered and air-dried to obtain AgCl/TiO₂ particles. 1.7–2 mL of TiCl₄ solution is added when preparing the red-light particles, and 1.3–1.5 mL of TiCl₄ solution is added when preparing the green-light particles.

Second, Functionalization of AgCl/TiO₂ particles. A certain amount of the prepared AgCl/TiO₂ particles are added into the EtOH to obtain a 50-mL suspension. The suspension is then transferred into a 100-mL three-necked flask and stirred at 90 rpm for 30 min. 2 mL of polyethylene glycol-400 (PEG-400, analytical reagent) is dissolved in 5 mL of EtOH and slowly dropped in the three-necked flask. After stirring the suspension for 1 h, 1 mL of γ -methacryloxy propyltrimethoxy silane (MPS, analytical reagent) is dissolved in 5 mL of EtOH and slowly added into the three-necked flask. Similarly, after stirring the suspension again for 5 h, 1 mL of ammonium hydroxide (25 wt%) is dissolved in 5 mL of EtOH and slowly dropped

into the three-necked flask. After being stirred for 10 h, the suspension is centrifuged at a rate of 2200 rpm for 3 min to discard the supernatant. The procedure is repeated 2 to 3 times, the precipitated MPS-functionalized AgCl/TiO₂ particles are obtained.

Third, PMMA-coated AgCl/TiO₂ particles (AgCl/TiO₂@PMMA). The AgCl/TiO₂@PMMA composite particles were synthesized by the route that the monomer was adsorbed onto the modified AgCl/TiO₂ followed by dispersion polymerization. A certain amount of the functionalized AgCl/TiO₂ particles is transferred to a 250 mL three-necked flask. 2 mL of methyl methacrylate (MMA, analytical reagent) and 10 μ L of ethylene glycol dimethacrylate (EGDMA, analytical reagent) are dissolved in 25 mL of EtOH. The mixture is then slowly dropped in the three-necked flask. After stirring the suspension in the three-necked flask at 90 rpm for 1 h, 0.2 g of polyvinyl pyrrolidone (PVP, analytical reagent) is dissolved in 80 mL of deionized water and added to the three-necked flask using a funnel. The suspension is continuously stirred for 1 h, and the three-necked flask is transferred to a thermostat water bath (80°C) and condensed with nitrogen. Subsequently, 0.06 g of kalium persulfate (KPS, analytical reagent) is dissolved in 6 mL of deionized water. Under constant stirring, 6 mL of KPS solution is added to the three-necked flask in three portions: 2 mL is dropped every 2 h. After the last addition of the KPS solution, the suspension is stirred for 6 h to complete the coating of AgCl/TiO₂ and obtain a suspension of AgCl/TiO₂@PMMA particles. The resulting suspension is centrifuged at 3000 rpm for 5 min to discard the supernatant. The remaining precipitate is then washed with the deionized water and centrifuged for several times. The final precipitate is washed with a small amount of deionized water before transferring to a 10 mL vial for storage.

Finally, Ag/AgCl/TiO₂@PMMA composite particles. The quartz glass is hydrophilically treated. The clean quartz glass (1 cm \times 2 cm) is sonicated in alcohol for 30 min, washed with deionized water, and then boiled for 1 h in a mixture of 30% hydrogen peroxide (H₂O₂, analytical reagent) and deionized water (7:3 by volume). The suspension of the AgCl/TiO₂@PMMA particles is spin-coated onto a hydrophilically treated glass substrate using a spin coater. Finally, the glass substrate coated with AgCl/TiO₂@PMMA particles is placed a photoreduction process under an incandescent lamp (or a xenon lamp, $\lambda > 420$ nm) for

10 h. Part of AgCl in the particles is decomposed into Ag elementary substance, which is precipitated on the surface of the particles to obtain Ag/AgCl/TiO₂@PMMA particles.”

The following is excerpted from the APPENDIX section of Ref. 46 (Zhao, J., et al. Broadband omnidirectional visible spectral metamaterials, *Photonics Res.* **11**, 1284-1293 (2023).):

“APPENDIX A: Preparation of Ag/AgCl/TiO₂@PMMA meta-cluster particles resonating in different bands of visible light.

Ag/AgCl/TiO₂@PMMA meta-cluster particles corresponding to red-light and green-light have been prepared using the solvothermal synthesis method. Subsequently, we prepared meta-cluster particles that resonate in different bands of visible light by fine-tuning the experimental conditions. The concentration of silver nitrate solution in the whole experimental system is 0.026 g/mL: With the increase of the dosage of TiCl₄ (in the range of 1.0 mL to 2.2 mL), the response wavelength of the prepared particles is red shifted. A variety of meta-cluster particles with response bands ranging from blue to red light spectrum (450 nm-680 nm) are prepared, covering most regions of visible light band. The concentration of silver nitrate solution in the whole experimental system is 0.0395 g/mL: Various meta-cluster particles with response bands ranging from 600nm to 700nm are prepared by adjusting the dosage of TiCl₄ (1.6mL to 2.5mL).

APPENDIX B: Preparation of Ag/AgCl/TiO₂@PMMA meta-cluster particles resonating in visible light broadband.

Inspired by polychromatic light, meta-cluster mixed particles that resonate in visible light broadband (480–700 nm) is prepared.

First, Preparation of meta-cluster particles resonating in the wavelength of 480–600 nm. Single frequency meta-cluster particle is dispersed in deionized water, and the volume ratio of the sample to deionized water is about 1:5. Then, 10 μL of the samples with transmission peaks of 540 nm, 570 nm, and 600 nm are taken, respectively, for full mixing, and the transmission peak is obtained at 500–600nm. Finally, 10 μL of the sample corresponding to 490 nm and 20 μL of the sample corresponding to 500nm are added into the above sample to fine tune the

transmission spectrum. Thus, the broadband sample with a transmission peak of 480–600 nm is obtained.

Second, Preparation of meta-cluster particles resonating in the wavelength of 600–700 nm.

The preparation method is similar to that described above. 10 μL of the samples with transmission peaks of 600 nm, 640 nm, 680 nm, and 700 nm are taken, respectively, for full mixing, and the broadband sample with a transmission peak of 600–700 nm is obtained.

Finally, Preparation of meta-cluster particles resonating in the wavelength of 480–700 nm.

The prepared samples that resonate in the wavelength of 480–600 nm and 600–700 nm are mixed in equal proportions. Then, broadband meta-clusters with transmission peak of 480–700 nm is obtained, by adding 10 μL of the sample corresponding to 700nm into the above mixed samples for fine tuning.”

And the following is the description in the Methods section of this manuscript:

“Preparation and characterisation of meta-clusters

The Ag/AgCl/TiO₂@PMMA meta-cluster particles corresponding to red-light and green-light are prepared using the solvothermal synthesis method⁴². In order to solve the problem of the coating of nano-silver layer of ball-thorn-shaped clusters, AgCl is firstly formed by mixing a certain amount of AgNO₃ into TiCl₄ during the process of preparing the TiO₂ rods. After a photoreduction method, AgCl further disintegrates into elemental chlorine and metallic silver. The latter precipitates on the outer surface of the ball-thorn-shaped structure to form the discrete silver distribution about 1 nm. Next, these agglomerated particles are immersed in PMMA and illuminated to form the Ag/AgCl/TiO₂@PMMA particles.

Eight kinds of Ag/AgCl/TiO₂@PMMA (meta-cluster) nanoparticles were prepared by solvothermal synthesis from bottom to top⁴⁶. The transmission electron microscopy (TEM) images of the meta-cluster particles that resonate in the red-, yellow-, and green- light spectra reveal a classic kernel (AgCl/TiO₂)–shell (PMMA) structure. The images confirm the presence of PMMA filling between different nanorods, and the thickness of the PMMA shell is nearly 20–30 nm. According to the idea of multi-frequency composite, the meta-cluster particle system in the wide band of visible light was prepared.

The meta-cluster particle set was characterized by the optical micrograph and scanning electron microscopy (SEM) images, the transmission electron microscopy (TEM) images, the

local high-angle annular dark-field imaging scanning TEM (HAADF-STEM) images, X-ray diffraction (XRD) patterns, Ultraviolet–visible– near infrared (UV-VIS-NIR) absorption spectra^{42, 46}.”

Our work is a series of consecutive studies, and the detailed preparation methods have been reported in previous studies, so the material preparation part is described briefly in this article. Our preparation methods are fully disclosed and, moreover, easily reproducible. The preparation process and materials provided are fully reproducible for colleagues engaged in bottom-up preparation. In addition, during the development of the samples, which took over five years, all samples were left for long periods of time and the performance remained stable and repeatable. The properties tested were repeated multiple times with reliable reproducibility.

Therefore, we have added the following description to the modified methods section “The reliability of the materials, including detailed procedures for sample preparation, can be found in the " **Methods**" section of Ref. 42 and the " **Appendix A, B, and C**" sections of Ref. 46, and the repeatability of the time stability and performance measurements have been repeatedly calibrated.”

REVIEWERS' COMMENTS

Reviewer #3 (Remarks to the Author):

I feel that the elements provided by the authors in their latest version address my last comments.

The elements presented seem to me to be sufficiently well described to facilitate reproduction of the results, highlighting the central role of previous developments by the same teams in constituting the guides studied in this paper.

The paper would probably benefit from reducing the sensationalist claims to focus on the factual elements supported by the proposed experimental data, which in their current form still don't seem to me to guarantee that the authors' assertions are the only possible explanation for the observed phenomena.

To sum up, I think this article presents original results despite the incremental nature of these developments.

After careful examination of the reviewers' comments, we have responded to the comments point by point (in red) in this letter. The followings are the responses in detail:

REVIEWERS' COMMENTS

Reviewer #3 (Remarks to the Author):

I feel that the elements provided by the authors in their latest version address my last comments.

The elements presented seem to me to be sufficiently well described to facilitate reproduction of the results, highlighting the central role of previous developments by the same teams in constituting the guides studied in this paper.

The paper would probably benefit from reducing the sensationalist claims to focus on the factual elements supported by the proposed experimental data, which in their current form still don't seem to me to guarantee that the authors' assertions are the only possible explanation for the observed phenomena.

To sum up, I think this article presents original results despite the incremental nature of these developments.

Response: Many thanks to the reviewer for recognizing our response and for acknowledging the results of the study. Regarding the interpretation of the observed phenomena, as with many phenomena, there can sometimes be different perceptions. As far as our current knowledge is concerned, this is a possible statement, and of course as more experimental results and related research results emerge, more explanations may be offered. We will make further studies in our continuing work.